# IgGM: A Generative Model for Functional Antibody and Nanobody Design

**Rubo Wang**[1,2,3,*] **Fandi Wu**[3*], **Xingyu Gao**[1,2,†] **Jiaxiang Wu**[3], **Peilin Zhao**[3], **Jianhua Yao**[3†]

[1]Institute of Microelectronics, Chinese Academy of Sciences, Beijing, China
[2]University of Chinese Academy of Sciences, Beijing, China
[3]Tencent AI Lab, Shenzhen, China

wangrubo@hotmail.com, gxy9910@gmail.com, {fandiwu, masonzhao, jianhuayao}@tencent.com

## ABSTRACT

Immunoglobulins are crucial proteins produced by the immune system to identify and bind to foreign substances, playing an essential role in shielding organisms from infections and diseases. Designing specific antibodies opens new pathways for disease treatment. With the rise of deep learning, AI-driven drug design has become possible, leading to several methods for antibody design. However, many of these approaches require additional conditions that differ from real-world scenarios, making it challenging to incorporate them into existing antibody design processes. Here, we introduce IgGM, a generative model for the de novo design of immunoglobulins with functional specificity. IgGM simultaneously generates antibody sequences and structures for a given antigen, consisting of three core components: a pre-trained language model for extracting sequence features, a feature learning module for identifying pertinent features, and a prediction module that outputs designed antibody sequences and the predicted complete antibody-antigen complex structure. IgGM effectively predicts structures and designs novel antibodies and nanobodies. This makes it highly applicable in a wide range of practical situations related to antibody and nanobody design. [1]

## 1 INTRODUCTION

Antibodies, also known as immunoglobulins (Ig), are Y-shaped proteins secreted by B lymphocytes, primarily found in blood and lymphatic fluid (Silverthorn, 2015; Akkaya et al., 2020). As shown in Figure 1(A), they consist of two heavy chains and two light chains, each containing a variable domain (VH or VL) and a constant domain (CH or CL). The variable regions include three complementarity-determining regions (CDRs), which are crucial for antigen binding and determine the antibody's specificity. Additionally, the variable regions contain four framework regions (FRs). The FRs provide structural support for the VR and exhibit relatively low variability. Antibodies play a critical role in the immune system by recognizing and binding to specific foreign substances such as bacteria, viruses, fungi, and parasites, and tagging them for clearance (Schroeder Jr & Cavacini, 2010; Litman et al., 1993). Their importance extends to medicine, scientific research, and biotechnology, where they are used in disease treatment, personalized medicine, vaccine development, and new drug development (Nelson et al., 2010; Weiner, 2015; Sliwkowski & Mellman, 2013).

Despite their significance, traditional methods for antibody production face challenges such as long production cycles (Georgiou et al., 2014), batch-to-batch variations (Bradbury et al., 2018), and the need for humanization (Safdari et al., 2013) to reduce immunogenicity. These challenges limit the widespread application and therapeutic efficacy of antibodies. To address these issues, researchers have turned to artificial intelligence for antibody design. Early approaches, such as energy-based computational methods (Li et al., 2014; Adolf-Bryfogle et al., 2018), were limited by the expressive capacity of statistical energy functions. Language models trained on sequences (Liu et al., 2020;

---

[*]Equal contribution.
[†]Corresponding author.

[1]Code is available at: https://github.com/TencentAI4S/IgGM

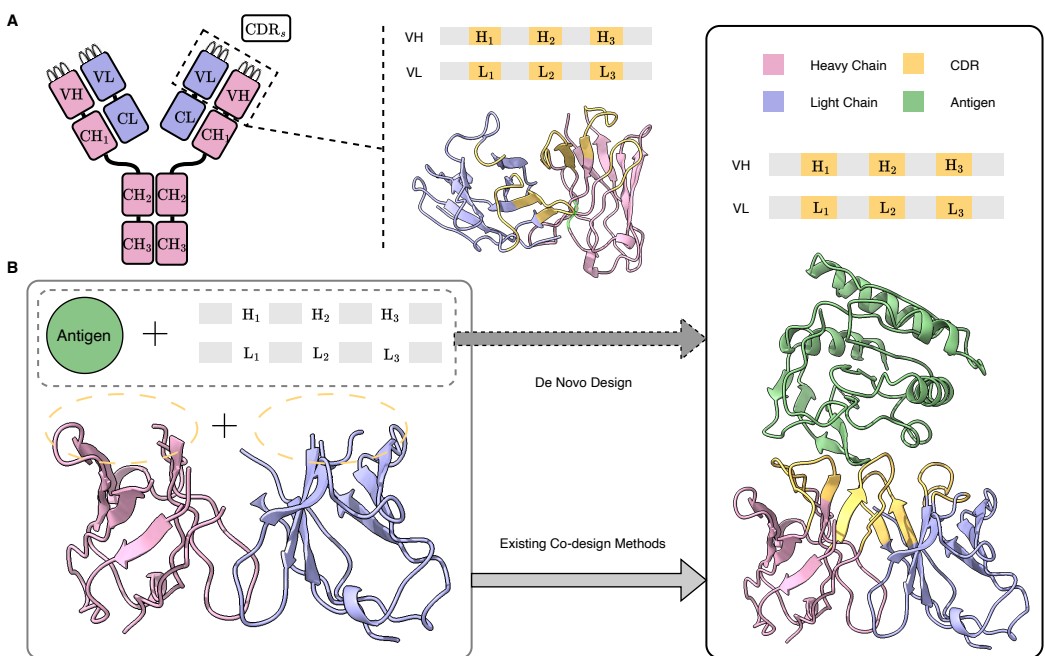

Figure 1: **(A)** An antibody consists of a symmetric Y-shaped structure, which includes variable regions (VH, VL) and constant regions (CH, CL). In practical antibody design, the focus is on the variable regions, which comprise the framework regions (FRs) and the complementarity-determining regions (CDRs). **(B)** De novo antibody design refers to the process of creating a novel antibody that can bind to a given antigen, where the framework regions can be selected based on sequences with favorable physicochemical properties. Existing co-design methods require the simultaneous provision of both the structure and sequence of the framework regions; however, in practical antibody design, the structures are often unknown.

Saka et al., 2021; Akbar et al., 2022; Shin et al., 2021; Jing et al., 2020; Cao et al., 2021) also showed suboptimal performance due to lack of structural knowledge. Recently, co-design methods that simultaneously design protein sequences and structures (Anishchenko et al., 2021; Wang et al., 2022; Anand & Achim, 2022; Shi et al., 2023) have demonstrated the feasibility of using AI for antibody design (Jin et al., 2021; 2022; Luo et al., 2022; Kong et al., 2023a;b; Wu & Li, 2024). Many existing co-design methods rely on known experimental structures of antibody-antigen complexes and the modification of existing antibodies. However, these structures and antibodies are not always available for the design of novel antibodies targeting a specific antigen. As shown in Figure 1(B), most approaches depend on actual framework region structures or templates derived from datasets. Unfortunately, such information is often missing when targeting a new antigen.

To overcome these limitations, we proposed IgGM, a generative model that performs simultaneous co-design of antibody sequence and structure. The overall process is shown in Figure 2. IgGM employs a multi-level network architecture. It first utilizes a pre-trained protein language model to extract evolutionary features of sequences. Then, a feature encoder studies the interactions between antigens and antibodies. Finally, a prediction module outputs the structures and sequences of the antibodies. IgGM leverages the interplay between sequence and structure to generate accurate antibody designs, even when only partial sequences of the framework region are available. This capability aligns with practical application scenarios and offers new possibilities for antibody design. IgGM excels at generating the CDR regions and their structures and can dock the generated structure to the corresponding epitope. It supports multiple design scenarios and can adapt to various conditions without the need for retraining, such as predicting antigen-antibody complexes, designing the CDR H3 region of antibodies, and designing multiple CDR regions. Furthermore, it can be extended to nanobodies, which are small single-domain antibodies that exhibit strong binding affinity to antigens and high stability (Cai et al., 2020). The experimental results indicate that IgGM achieves superior performance in multiple design tasks, demonstrating accuracy in structure prediction tasks that is

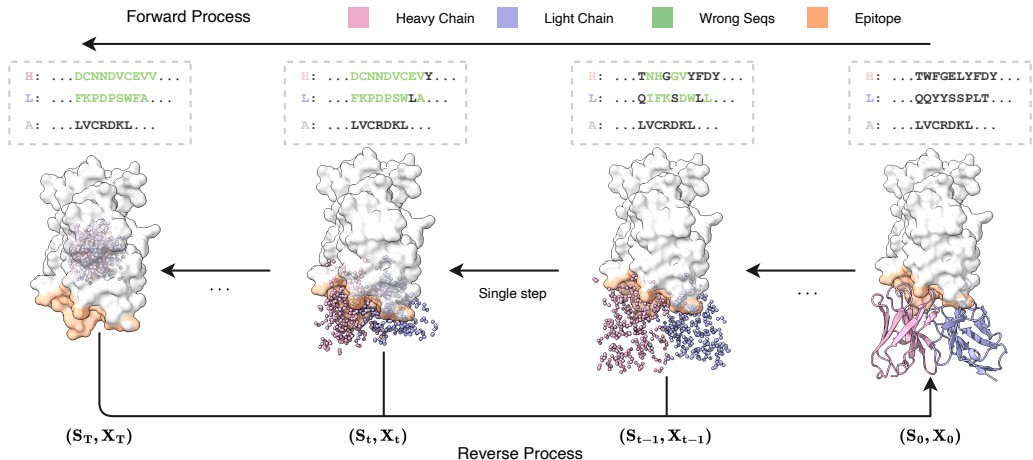

Figure 2: The forward process and reverse process for Co-Design of Antibody Sequences and Structures: The figure illustrates an example trajectory in the design of antibodies, incorporating specific antigen and antibody framework region sequences. It depicts the input data $(s_T, x_T)$ to the model, along with the corresponding predicted outcomes $(s_0, x_0)$. The consistency model is capable of generating antibody sequences and structures across diverse noise levels, demonstrating the gradual refinement from noisy inputs to a well-defined antibody structure.

comparable to existing structure prediction methods. In the Appendix G, we discussed the future research directions and existing limitations.

## 2 BACKGROUND

### 2.1 PRELIMINARIES

Since nanobodies can be considered as a single heavy chain of an antibody, we will use antibodies as an example in the following discussion. Proteins are composed of 20 different amino acids. For a given protein of length $N$, the protein sequence can be represented as $\mathcal{S} = \{s_i\}_{i=1}^N$, where each $s_i$ denotes a residue. The three-dimensional structure of the protein can be represented by the three-dimensional coordinates of the backbone atoms, denoted as $\mathcal{X} = \{x_{i,\omega}\}_{i=1}^N$, where each $x_{i,\omega} \in \mathbb{R}^3$ and $\omega \in \{C\alpha, N, C, O\}$. Antibodies are specialized proteins that consist of two distinct chains, while nanobodies contain only a single heavy chain. Each chain is composed of four framework regions (FRs) and three complementarity-determining regions (CDRs), The CDRs of each chain can be further divided into three segments: CDR H1, H2, H3 for the heavy chain, and CDR L1, L2, L3 for the light chain. An antigen-antibody complex can be represented as

$$\mathcal{H} : \mathcal{L} - \mathcal{A} = \{(s_i, x_{i,\omega}) | i \in \{1, ..., l_{\mathcal{H}}, ..., l_{\mathcal{H}} + l_{\mathcal{L}}, ..., l_{\mathcal{H}} + l_{\mathcal{L}} + l_{\mathcal{A}}\}\}, \quad (1)$$

where $\mathcal{H} : \mathcal{L} - \mathcal{A}$ represents the heavy chain and light chain of the antibody, as well as the chain of the antigen, $l$ represents the length of the amino acid sequence for each chain. In the case of a nanobody-antigen complex, the light chain is not included.

Previous studies have defined the antibody design problem as selecting a framework region (Shin et al., 2021; Akbar et al., 2022) to design the CDRs to create antibodies that can bind to specific antigens. Since the influence of the framework region on the antigen-antibody interaction is relatively minor, earlier research has primarily focused on designing the CDRs, often assuming that the structure of the framework region is fixed and unchanging. However, when faced with a completely new antigen, the structure of the resulting antibody, including the framework region, is unknown and cannot be predetermined. IgGM considers structural changes in the framework regions during the binding process, enabling the design of the whole antibody structure even without experimental structures. Given the role of framework regions in providing support and certain sequences having favorable physicochemical properties for practical reuse (Bennett et al., 2024; Vincke et al., 2009),

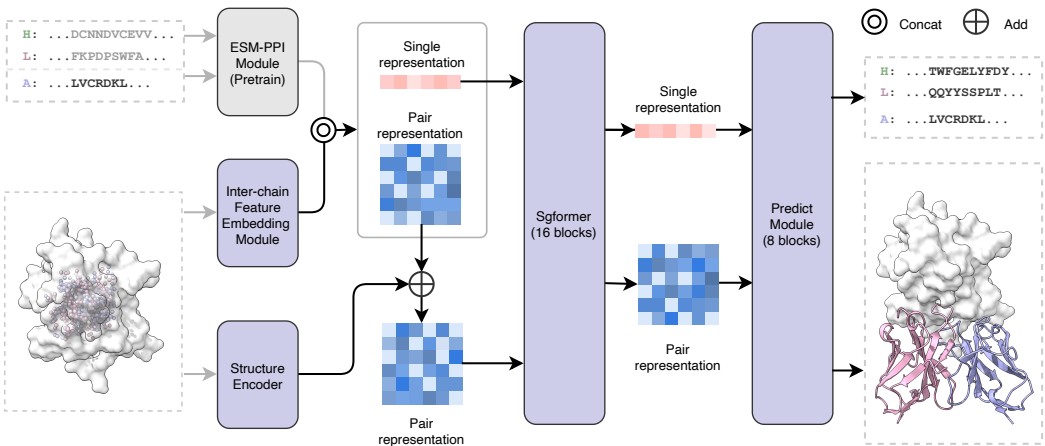

Figure 3: IgGM Model Framework Diagram: Before being input into the model, the antibody sequence and structure are pre-noised. The model includes a pre-trained ESM-PPI module, a Inter-chain Feature Embedding Module, a Structure Encoder, the Feature Embedding Module, 16 layers of Sgformer, and an 8 layers of Prediction Module.

there's no necessity to design entirely new framework region sequences. Therefore, our focus remains on designing CDR sequences.

## 2.2 PROBLEM FORMULATION

In practical antibody design, existing work has utilized the sequences of known framework regions to guide antibody design (Bennett et al., 2024). However, in the absence of the complementarity-determining region (CDR), the structure of the antibody is not fixed. Therefore, we investigate the problem in real-world application scenarios, specifically designing antibody structures and sequences that can bind to specific locations (epitopes) on the antigen, given the framework region sequences. We separate the residues of CDRs and the framework regions for easier identification. The set of residues contained in the CDRs is given by:

$$\mathcal{R}_C := \left\{ (\mathcal{S}_{\mathrm{C}}, \mathcal{X}_{\mathrm{C},\omega}) | (\mathcal{S}_{\mathrm{C}}, \mathcal{X}_{\mathrm{C},\omega}) \in \{(\mathcal{S}_{\mathrm{CDRs}}, \mathcal{X}_{\mathrm{CDRs}})\} \right\}, \tag{2}$$

the set of residues representing the framework regions is denoted as

$$\mathcal{R}_{\mathrm{F}} := \left\{ (\mathcal{S}_{\mathrm{F}}, \mathcal{X}_{\mathrm{F},\omega}) | (\mathcal{S}_{\mathrm{F}}, \mathcal{X}_{\mathrm{F},\omega}) \in \{(\mathcal{S}_{\mathrm{FRs}}, \mathcal{X}_{\mathrm{FRs}})\} \right\}. \tag{3}$$

The entire design problem can be simplified to the task of designing the sequences of CDRs $s_{\mathrm{C}}$ and the overall structure $(\mathcal{X}_{\mathrm{C},\omega}, \mathcal{X}_{\mathrm{F},\omega})$ of the antibody, given the sequence and structure of the antigen $\mathcal{R}_{\mathrm{A}}(\mathcal{S}_{\mathrm{A}}, \mathcal{X}_{\mathrm{A},\omega})$ as well as the sequences of the antibody's framework regions $\mathcal{S}_{\mathrm{F}}$.

## 3 METHODS

We introduce IgGM, a versatile antibody design model suitable for designing antibodies that can bind to specific antigens. The model aims to facilitate the flexible design of various tasks related to antibody sequences and structures. We first discuss the denoisng network architecture in Section 3.1, followed by the training methods and objectives of IgGM in Section 3.2, and finally, we address the sampling methods in Section 3.3.

## 3.1 DENOISING NETWORK ARCHITECTURE

The overall network architecture is illustrated in Figure 3, which includes a pre-trained protein language model, a multi-level feature encoder, and a sequence and structure design module. Given an antigen structure and an initialized sampled antibody sequence, the pre-trained protein language model first extracts features from the sequence, which encompass evolutionary information about the

protein. These features are then fused through a feature encoder (Sgformer), and finally, a sequence and structure design module (prediction module) is employed to generate the antibody sequence and the structure that binds to the antigen.

**Feature extraction from protein language models.** Inspired by the success of pre-trained language models in natural language processing, we employ pre-trained protein language models (Lin et al., 2022; Chen et al., 2023; Hayes et al., 2024) as a feature extractor. We select ESM-PPI (Wu et al., 2024) as our sequence feature extractor due to its ability to handle inter-chain relationships. ESM-PPI is an extension of the ESM2 model (Lin et al., 2022), which has been further refined to improve its proficiency in capturing the structural and functional characteristics of multi-chain protein complexes. The antigen and perturbed antibody data are processed by the PLM, with the features from the final layer being meticulously extracted to serve as input to feature encoder. To preserve the integrity of the learned features and to conserve computational resources, we maintain the PLM's parameters in a frozen state throughout this process. The enhanced capabilities of ESM-PPI for multi-chain protein structure prediction are detailed in (Wu et al., 2024), where readers can find a more in-depth discussion of the model's architecture, training procedure, and its application.

**Multi-level feature encoder.** To fully leverage the interactions between different features, we utilize a multi-level feature encoder for feature encoding, as illustrated in Figure 3. This approach allows the model to develop an understanding of the distinct chains that comprise the antibody structure. We incorporate chain-specific representations into the output features of the pre-trained language model (PLM). Furthermore, to address the critical aspect of antigen epitopes, we augment the antigen feature set with specialized representations that emphasize the interactions between the antibody and the antigen. We employ a structural encoder to provide the model with a means to identify the precise positions of individual amino acids. Subsequently, these extracted features are input into Sgformer consisting of 16 blocks for further feature fusion and encoding. The sequence features extracted at this stage are pivotal for the subsequent recovery of the original sequence from the perturbed input, while the pair-wise representations encode the relational information essential for understanding the complex folding of the antibody-antigen complex.

**Sequence and structure design module.** IgGM employs 8 layers of Predict modules, as illustrated in Figure 6. The Predict modules utilize invariant point attention to optimize the structure while simultaneously outputting the designed sequences. Due to the invariance of the Predict modules, this ensures that the model's predictions remain consistent regardless of the orientation or position of the antibody in space. The Predict modules fully leverage the sequence features and pair-wise representations learned by the Sgformer modules, while also incorporating the structures obtained from the initial sampling as input. By integrating these features with the invariant point attention mechanism, the Predict modules are able to iteratively refine the coordinates of the amino acids, ultimately revealing the precise spatial arrangement of the antibody's three-dimensional structure.

**Inter-chain Feature Embedding Module and Structure Encoder.** IgGM utilizes two components to leverage the distinct characteristics of different chains and epitope information, as illustrated in Figure 3. The Inter-chain Feature Embedding Module integrates positional information of amino acids and inter-chain information to fuse features, thereby capturing the distinct positional characteristics of the chains while also obtaining chain-specific features. The Structure Encoder primarily encodes the protein structure; this module employs distance information to derive spatial features between pairs of amino acids, converting them into features through a discretization process. To effectively utilize epitope information, we implement a specialized processing approach. Specifically, we encode the sequence epitope and spatial contact information into Single representation and Pair representation, respectively, to facilitate the effective generation of structures near the epitope. As shown in the Figure 8, IgGM can design antibodies that specifically bind to designated epitopes.

## 3.2 TRAINING DETAILS

We train our model on a structural dataset using a distillation approach to train a consistency model. First, we pre-train a diffusion model, which consists of two phases. Ablation studies (Appendix E) demonstrate that this two-phase training approach is crucial for the successful training of the model. In the first phase, we focus on training the structural component while preserving the information of the original sequences, specifically by conducting training tasks solely for structure prediction. During the training process, we sample a pair of antigen-antibody complexes $x$ from the dataset

$\{s, x\}$ and add noise at different time steps. We randomly select a time step $t$ to introduce noise, resulting in $x_t$. The model $\mathcal{D}$ is then trained to recover the overall antibody structure. Our objective is to ensure that the recovered structure closely resembles the true structure. For the protein structure, we utilize a combined loss function. Below, we provide a brief introduction, and for more detailed information, please refer to the Appendix D. The overall loss is as follows:

$$\mathcal{L} = \mathcal{L}_{\text{geo}} + \mathcal{L}_{\text{Frame}} + \mathcal{L}_{\text{iFrame}} + 0.02\mathcal{L}_{\text{viol}}. \tag{4}$$

Here, $\mathcal{L}_{\text{geo}}$ is designed to provide more direct supervision in the subsequent stack. Four auxiliary heads, implemented as feed-forward layers, are added to the top of the final pair features to predict inter-residue distances and angles, as described in trRosetta (Yang et al., 2020). The term $\mathcal{L}_{\text{Frame}}$ is intended to provide direct supervision in the prediction module to recover the antibody structure, as proposed in RFDiffusion (Watson et al., 2023), and we extend it to multi-chain scenarios as $\mathcal{L}_{\text{iFrame}}$ to enhance the model's focus on the structural stability of the binding site. The formula can be expressed as follows:

$$L_{\text{Frame}} = \frac{1}{\sum_{i=0}^{I-1} \gamma^i} \sum_{i=1}^{I} \gamma^{I-i} d_{\text{Frame}}(x^{(0)}, \hat{x}^{(0),i})^2, \tag{5}$$

$$L_{\text{iFrame}} = \frac{1}{\sum_{i=0}^{I-1} \gamma^i} \sum_{i=1}^{I} \gamma^{I-i} d_{\text{iFrame}}(x^{(0)}, \hat{x}^{(0),i})^2, \tag{6}$$

where $i$ represents the output of the $i$-th layer of the structural module, while $\gamma$ can increase the weights of the subsequent layers. The term $\mathcal{L}_{\text{viol}}$ serves as a penalty term to correct incorrect bond lengths, bond angles, and spatial conflicts, as introduced in AlphaFold-Multimer (Evans et al., 2021). Notably, we do not penalize the bond length and angle between the last residue in the heavy chain and the first residue in the light chain, as there is no peptide bond between them.

$$\mathcal{L}_{\text{viol}} = \mathcal{L}_{\text{bond-length}} + \mathcal{L}_{\text{bond-angle}} + \mathcal{L}_{\text{clash}}. \tag{7}$$

Next, we move into the second stage of training, expanding on the groundwork laid in the initial phase by focusing on sequence design. The model is trained using the following objective:

$$\mathcal{L} = \mathcal{L}_{\text{srcv}} + \mathcal{L}_{\text{geo}} + \mathcal{L}_{\text{Frame}} + \mathcal{L}_{\text{iFrame}} + 0.02\mathcal{L}_{\text{viol}}. \tag{8}$$

Here, $\mathcal{L}_{\text{srcv}}$ is designed to supervise the model in recovering the amino acid sequence using a cross-entropy loss function. This loss function encourages the model to predict the correct amino acid probabilities for each position in the designed sequence, thereby enhancing the overall performance of the sequence design. It is noteworthy that during the second phase of training, we employed a mixed training approach. In the model training process, for the sequences, we assigned probabilities of $4 : 2 : 2 : 2$ for the model to design CDR H3, CDR H, and all CDRs, due to the greater variability observed in the heavy chain CDRs. Ablation studies (Appendix E) indicate that this approach effectively enhances the performance of IgGM and equips the model with the capability to design various regions, such as predicting structures and designing sequences for all CDR regions.

After completion of the diffusion model training, we employed distillation training to obtain the final consistency model. We followed Song et al. (2023) training methodology, aiming to minimize the consistency distillation loss:

$$\mathcal{L}_{\text{CD}}\left(\boldsymbol{\theta}, \boldsymbol{\theta}^-; \Psi\right) = \mathbb{E}_{\boldsymbol{z}, \boldsymbol{c}, n}\left[d\left(\boldsymbol{f_\theta}(\boldsymbol{z}_{t_{n+1}}, \boldsymbol{c}, t_{n+1}), \boldsymbol{f_{\theta^-}}(\hat{\boldsymbol{z}}_{t_n}^\Psi, \boldsymbol{c}, t_n)\right)\right]. \tag{9}$$

---

**Algorithm 1** IgGM Sampling

---

**Input:** Model $\boldsymbol{f_\theta}(\cdot, \cdot)$, sequence of time points $\tau_1 > \tau_2 > \cdots > \tau_{N-1}$, initial noise $(\hat{\mathbf{s}}_T, \hat{\mathbf{x}}_T)$, antigen $\mathbf{s}_A, \mathbf{x}_A$)
$(\mathbf{s}, \mathbf{x}) \leftarrow \boldsymbol{f_\theta}((\hat{\mathbf{s}}_T, \hat{\mathbf{x}}_T), T, (\mathbf{s}_A, \mathbf{x}_A))$
**for** $n = 1$ **to** $N - 1$ **do**
    Sample $\bar{Q}_{\mathbf{z}} = Q_1 Q_2 ... Q_T \sim q(x_t = j | x_{t-1} = i), \quad \mathbf{x_z} \sim (\mathcal{N}(\mathbf{0}, \boldsymbol{I}), \text{Uniform}(SO(3)))$
    $\hat{\mathbf{s}}_{\tau_n} \leftarrow \mathbf{s}\bar{Q}_{\mathbf{z}}, \quad \hat{\mathbf{x}}_{\tau_n} \leftarrow \mathbf{x} + \sqrt{\tau_n^2 - \epsilon^2}\mathbf{x_z}$
    $\mathbf{x} \leftarrow \boldsymbol{f_\theta}((\hat{\mathbf{s}}_{\tau_n}, \hat{\mathbf{x}}_{\tau_n}), \tau_n, (\mathbf{s}_A, \mathbf{x}_A))$
**end for**
**Output:** $(\mathbf{s}, \mathbf{x}) = 0$

---

In this context, $\hat{z}_{t_n}^{\Psi}$ represents an estimate of the evolution of the PF-ODE from $t_{n+1}$ to $t_n$ utilizing the ODE solver $\Psi$:

$$\hat{z}_{t_n}^{\Psi} - z_{t_{n+1}} = \int_{t_{n+1}}^{t_n} \left( f(t)z_t + \frac{g^2(t)}{2\sigma_t} \epsilon_\theta\left(z_t, c, t\right) \right) dt \approx \Psi(z_{t_{n+1}}, t_{n+1}, t_n, c), \tag{10}$$

where the integration from $t_{n+1}$ to $t_n$ is approximated using the solver $\Psi(\cdot, \cdot, \cdot, \cdot)$.

## 3.3 DIRECTLY GENERATE ANTIBODIES

As shown in Algorithm 1, when designing antibodies against a specific antigen, we first randomly sample one type of amino acid from the 20 standard amino acids to serve as the initial amino acid. We then sample the translation coordinates from a Gaussian distribution and the initial rotation from the standard SO(3) group. These initial variables are subsequently sampled and generated using a trained model. Due to the characteristics of the consistency model, it is capable of recovering real data from different time points. Therefore, we can also replace the initial coordinates, for instance, by using structural prediction tools such as AlphaFold3 to initialize the structure, thereby achieving higher quality antibody generation. During the generation process, the advantages of the consistency model allow IgGM to either generate in a single step or optimize through multi-step sampling to enhance the stability of the generated results. In single-step sampling, taking the collaborative design of sequences and structures as an example, once the initial sequence and structure are obtained from noise, the model can generate the final sequence and structure in one step. In contrast, multi-step sampling enables the generation of relatively more stable structures and sequences. We compared the performance differences between multi-step and single-step generation in Appendix Table 4, where we selected 10 steps to achieve a balance between quality and speed. For proteins, structure determines function, and this is particularly true for antibodies, where the stability of binding is crucial for effective interaction (Majewski et al., 2019). In this context, we have chosen the DockQ score as our selection criterion, as it measures the quality of interactions. The DockQ score provides a comprehensive assessment of the binding interface, taking into account factors such as the accuracy of the predicted complex structure and the stability of the interaction.

## 4 EXPERIMENTS

We constructed our training, validation, and test sets from the SAbDab database, employing the widely used method of dividing the dataset based on time, as previously established in other works (Jumper et al., 2021; Ruffolo et al., 2023; Wu et al., 2024; Abramson et al., 2024). We removed antibodies from the second half of 2023 that exhibited high sequence similarity to those in the training set to construct the test set, resulting in a test set comprising 60 antibodies (SAb-23H2-Ab) and a test set of 27 nanobodies (SAb-23H2-Nano). More details can be found in the Appendix C.1. Due to the limitations of AlphaFold 3, we generate five samples for each example to ensure a fair comparison. The details of the evaluation metrics can be found in Appendix C.4. Moreover, an ablation study of IgGM is in Appendix E.

Table 1: Complex structure prediction. Methods with superscript use antigen structure as input, while methods with superscript * utilize epitope information as input; methods with superscript use multiple sequence alignment (MSA). (AF3) indicates that the structure predicted by AlphaFold 3 is used as the initial input. The root mean square deviation (RMSD) in CDR H3 is reported. **Bold** indicates the best performance, while underline indicates the second-best performance.

| Method | Antibody Structure | | | Docking | | | |
|---|---|---|---|---|---|---|---|
| | TM-Score↑ | lDDT↑ | RMSD↓ | DockQ↑ | iRMS↓ | LRMS↓ | SR↑ |
| IgFold[†]→HDock | 0.9577 | 0.9019 | 2.1715 | 0.0218 | 16.6519 | 48.1571 | 0.0000 |
| tFold-Ag[*‡] | 0.9634 | 0.9142 | 1.9489 | 0.2522 | 6.7957 | 21.0346 | 0.4068 |
| AlphaFold 3[‡] | **0.9729** | **0.9305** | **1.5063** | 0.2951 | 10.9645 | 32.4080 | 0.3684 |
| dyMEAN[†*] | 0.9572 | 0.8882 | 2.2521 | 0.1005 | 8.9227 | 27.4234 | 0.0667 |
| IgGM[†*] | 0.9591 | 0.8956 | 2.1997 | 0.2986 | 6.2195 | 19.4888 | 0.4667 |
| IgGM[†*](AF3) | 0.9580 | 0.8941 | 2.1422 | **0.3630** | **3.8635** | **11.2647** | **0.6667** |

## 4.1 COMPLEX STRUCTURE PREDICTION

Complex structure prediction involves predicting the structure of the complex based on a given antibody sequence and antigen. IgGM can achieve structure prediction without the need for sequence design. Following the evaluation criteria of tFold-Ag (Wu et al., 2024), we assess the results using the TM-Score (Zhang & Skolnick, 2004), DockQ (Basu & Wallner, 2016), and success rate(DockQ> 0.23) of the predicted overall structure of the complex. We evaluated the structural prediction performance on the SAb23H2 test set, comparing three methods: the antibody structure prediction methods IgFold (Ruffolo et al., 2023) and tFold-Ag (Wu et al., 2024), as well as the latest protein structure prediction method AlphaFold 3 (Abramson et al., 2024), alongside the antibody design method dyMEAN (Kong et al., 2023b). Since IgFold only supports the prediction of antibody structures, we utilized HDock (Yan et al., 2020) to dock the predicted antibody structures with the original antigen. For tFold-Ag and AlphaFold 3, which can directly predict the structure of antigen-antibody complexes, we used the antigen-antibody sequences as input. In the case of dyMEAN and IgGM, we predicted the structure of the antigen-antibody complex given the antigen, using the antibody sequence as input.

As shown in Table 1, for the predicted antibody structures, IgGM outperformed dyMEAN, which uses templates for initialization, in terms of antibody structure prediction. Although there is still a gap compared to specialized structure prediction methods, the results are overall quite close, indicating that our method has learned the distribution of antibody structures. In terms of docking performance, our method surpassed both structure prediction methods and antibody design methods on DockQ, demonstrating that IgGM can achieve high docking performance. Additionally, it showed improved accuracy on iRMS and LRMS, with a success rate of 0.4667, significantly higher than dyMEAN's 0.067. This indicates that IgGM is capable of capturing the interactions between antigens and antibodies effectively. When using the structures predicted by AlphaFold 3 as the initial input for IgGM instead of randomly initialized structures, IgGM demonstrated improved performance across all metrics, particularly in docking-related indicators. Specifically, utilizing AlphaFold 3 structures increased the success rate by 20% compared to using randomly initialized structures. As shown in Figure 9, for the inaccurately predicted structures by AlphaFold3, IgGM is capable of making corrections to yield more suitable structures.

## 4.2 DE NOVO DESIGN OF ANTIBODIES FOR SPECIFIC ANTIGEN

Due to the limitations of previous methods in achieving end-to-end antibody design, we employed two pipelines to evaluate these approaches. Specifically, the first pipeline follows the dyMEAN process (structure prediction⇒docking⇒CDR generation⇒side-chain packing). For a given antibody sequence, we use IgFold (Ruffolo et al., 2023) to predict the antibody structure, followed by docking with the antigen using HDock (Yan et al., 2020), and then apply design methods for CDR generation. The second pipeline utilizes AlphaFold 3 (Abramson et al., 2024) for complex structure prediction (replacing IgFold⇒HDock), after which the antigen structure is aligned, and the predicted antigen structure is replaced with the native antigen structure to ensure that the antigen structure does not influence subsequent evaluations. In terms of antibody design, we evaluated several methods, including MEAN (Kong et al., 2023a), which employs graph neural networks to simultaneously generate sequences and structures for CDR H3; DiffAb (Luo et al., 2022), which uses diffusion models to generate sequences and structures for the CDR regions of antibodies; and dyMEAN (Kong et al., 2023b), which utilizes an end-to-end model for antibody design, allowing for novel structural designs through template utilization.

We tested various methods for the collaborative design of antibody sequence structures, with results presented in Table 2. It is noteworthy that MEAN (Kong et al., 2023a) is limited to designing the CDR H3 region. Therefore, we only tracked the performance of these two methods in this specific region. In contrast, DiffAb (Luo et al., 2022), dyMEAN (Kong et al., 2023b), and IgGM are capable of simultaneously designing all six CDR regions in both the light and heavy chains of antibodies. IgGM outperformed all other methods across nearly all metrics, such as sequence recovery rate and accuracy of generated docking positions, demonstrating the effectiveness of our approach. In comparison to DiffAb and MEAN, which do not support the design of overall structures, their performance varied when using different structures for initialization. The structures predicted by AlphaFold 3 were more accurate, leading to better docking position accuracy than dyMEAN as shown in Figure 4(A), although still lower than that of IgGM. Notably, IgGM was the only method with

Table 2: Results of the novel antibody design on SAb-2023H2-Ab. (IgFold) or (AF3) indicates that the antibody structure predicted by IgFold or AlphaFold 3 is used as the initial input. Backbone RMSD in different CDR regions are reported. H1-H3 indicate the CDRs of heavy chain while L1-L3 indicate the CDRs of light chain. **Bold** indicates the best performance, while underline represents the second best.

| Model | | DiffAb (IgFold) | DiffAb (AF3) | MEAN (IgFold) | MEAN (AF3) | dyMEAN | IgGM | IgGM (AF3) |
|---|---|---|---|---|---|---|---|---|
| AAR↑ | L1 | 0.597 | 0.608 | - | - | 0.633 | **0.750** | 0.737 |
| | L2 | 0.598 | 0.599 | - | - | 0.634 | **0.743** | 0.735 |
| | L3 | 0.421 | 0.424 | - | - | 0.570 | **0.635** | 0.602 |
| | H1 | 0.642 | 0.637 | - | - | **0.742** | 0.740 | 0.739 |
| | H2 | 0.363 | 0.394 | - | - | 0.627 | **0.644** | 0.639 |
| | H3 | 0.214 | 0.226 | 0.248 | 0.246 | 0.294 | **0.360** | 0.330 |
| RMSD↓ | L1 | 0.783 | 0.749 | - | - | 0.864 | **0.589** | 0.659 |
| | L2 | 0.471 | 0.466 | - | - | 0.481 | **0.378** | 0.395 |
| | L3 | 1.002 | 1.017 | - | - | 0.941 | **0.847** | 0.903 |
| | H1 | 0.650 | 0.623 | - | - | 0.633 | **0.555** | 0.590 |
| | H2 | 0.641 | 0.586 | - | - | 0.705 | **0.486** | 0.566 |
| | H3 | 2.741 | 2.646 | 2.357 | 2.300 | 2.454 | **2.131** | 2.155 |
| Docking | DockQ↑ | 0.022 | 0.208 | 0.022 | 0.207 | 0.079 | 0.246 | **0.326** |
| | iRMS↓ | 17.034 | 9.731 | 16.838 | 8.968 | 9.698 | 6.579 | **4.030** |
| | LRMS↓ | 48.163 | 27.559 | 48.104 | 27.557 | 28.764 | 19.678 | **11.229** |
| | SR↑ | 0.000 | 0.368 | 0.000 | 0.354 | 0.049 | 0.433 | **0.627** |

an iRMS below 8 and an LRMS below 20, while achieving a docking success rate of 43.3%, highlighting its generalizability. In terms of sequence and structural fidelity in the CDR regions, IgGM achieved a higher sequence recovery rate, with a 36% recovery rate in the highly flexible CDR-H3 region, representing a 22.4% improvement over the previous state-of-the-art method, dyMEAN. Additionally, the generated structures exhibited smaller deviations from the true structures. We also implemented a method using structures predicted by AlphaFold 3 as the initial input. The generated results demonstrate a significant improvement in docking-related metrics, with an increase in success rate of nearly 20%. Although there was a slight decline in performance regarding the CDR sequences and structures of the antibodies, the decrease was minimal. This indicates that utilizing powerful structural prediction methods to pre-estimate structures can effectively enhance the quality of the designed structures.

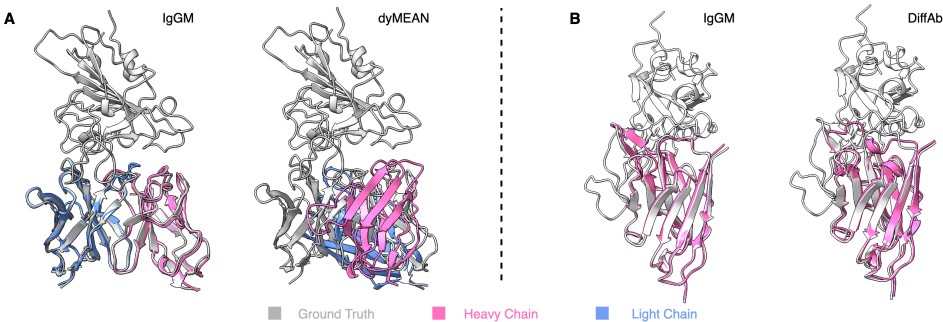

Figure 4: **(A)** Structures of the designed antibodies (PDB: 8hpu), designed by IgGM (left, DockQ = 0.824) and dyMEAN (right, DockQ = 0.029). **(B)** Structures of the designed nanobodies (PDB: 8q93), designed by IgGM (left, DockQ = 0.766) and DiffAb (right, DockQ = 0.495).

Table 3: Results of structure prediction and novel nanobody design on SAb-2023H2-Nano. (AF3) signifies that the structure predicted by AlphaFold 3 is utilized as the initial input. Backbone RMSD in different CDR3. CDR1-CDR3 indicate AAR of the rigons. **Bold** indicates the best performance.

| Structure prediction of Nanobody | | | | | | | | |
|---|---|---|---|---|---|---|---|---|
| Method | TM-Score↑ | lDDT↑ | RMSD↓ | | DockQ↑ | iRMS↓ | LRMS↓ | SR↑ |
| tFold-Ag | 0.9344 | **0.9303** | 1.6722 | | 0.2881 | **6.3490** | **15.0810** | 0.4296 |
| AlphaFold 3 | **0.9519** | 0.9286 | **1.1885** | | 0.2867 | 11.2194 | 32.6760 | 0.3885 |
| IgGM | 0.9318 | 0.8931 | 1.9925 | | **0.2907** | 7.9879 | 22.0168 | **0.4400** |
| Design of Nanobody | | | | | | | | |
| Method | CDR1↑ | CDR2↑ | CDR3↑ | RMSD↓ | DockQ↑ | iRMS↓ | LRMS↓ | SR↑ |
| DiffAb (AF3) | 0.533 | 0.291 | 0.156 | 2.274 | 0.211 | 13.265 | 35.805 | 0.346 |
| IgGM | **0.565** | **0.330** | **0.183** | **1.980** | **0.267** | **6.927** | **14.966** | **0.415** |

### 4.3 STRUCTURE PREDICTION AND DE NOVO DESIGN OF NANOBODIES

Nanobody sources are a type of single-domain antibody known as VHH fragments (Harmsen & De Haard, 2007). Nanobodies offer several significant advantages over traditional antibodies. Their structure is simpler, yet they possess a longer CDR3 region, making them particularly suitable for modifications and fusions with other proteins or biomolecules, thereby enabling the creation of multifunctional therapeutic and diagnostic agents based on nanobodies. Thanks to the extensibility of IgGM, it can also be used to design nanobodies. We evaluated the performance of IgGM in predicting nanobody structures using the SAb-2023H2-Nano dataset, comparing it with DiffAb and using structures predicted by AlphaFold 3 as the initialization.

As shown in Table 3, we assessed IgGM in terms of structural prediction and novel design for nanobodies. In the structural prediction of nanobody complexes, the relevant metrics for nanobody structures show an overall improvement compared to traditional antibodies. This enhancement is due to the fact that nanobodies consist of a single chain, making them relatively easier to analyze than the two-chain structure of conventional antibodies. However, for docking-related metrics, the performance of nanobodies is somewhat inferior to that of antibodies. This decline is attributed to the more flexible binding modes of nanobodies, which complicate the accurate prediction of the correct binding positions. IgGM achieved the best performance in terms of success rate and DockQ metrics, demonstrating a success rate of 44%. For nanobody design, IgGM outperformed DiffAb (AF3) across various metrics; however, there was a noticeable decline in the sequence recovery rate of the CDR regions compared to Antibody. This decline can be attributed to the longer CDR regions of nanobodies, which complicate the design process. In terms of structure, IgGM surpassed DiffAb (AF3), benefiting from the robust structural prediction capabilities of AlphaFold 3 as shown in Figure 4(B). While DiffAb (AF3) achieved a success rate of 0.346, the structures predicted by IgGM were more accurate, resulting in a success rate of 0.415.

## 5 CONCLUSION

In this study, we introduce IgGM, a generative model for antibody design that leverages consistency models to jointly design CDR sequences and the entire antibody structure. Unlike conventional approaches that target specific regions, such as CDR H3, IgGM considers the whole antibody, requiring only the target antigen and an antibody framework sequence. By integrating structural data, IgGM enhances specificity and quality, resulting in higher success rates for predicting binding positions. Our experimental results demonstrate that IgGM outperforms traditional methods in terms of accuracy and efficiency. Furthermore, IgGM holds potential in the design of nanobodies and has the ability to achieve the design of CDRs with specific lengths. With the accumulation of experimental data and advancements in structural prediction methodologies, IgGM's capabilities are anticipated to be substantially augmented, thereby establishing it as a formidable instrument for practical antibody design and expediting the development and application of antibodies.

ACKNOWLEDGMENTS

This work was supported in part by Science and Technology Innovation (STI) 2030—Major Projects under Grant 2022ZD0208700, and National Natural Science Foundation of China under Grant 62376264.

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

## A  RELATED WORK

**Protein design.**  Protein design aims to create new protein molecules with multiple functions (Dahiyat & Mayo, 1997). Traditional methods utilize energy functions, such as molecular dynamics simulations, for protein design; however, these energy function-based approaches are time-consuming and not suitable for practical protein design. In recent years, various machine learning methods have been applied to protein design, focusing on either sequences (Madani et al., 2023; Ferruz et al., 2022; Munsamy et al., 2022; Alamdari et al., 2023) or structures (Watson et al., 2023; Ingraham et al., 2023; Lutz et al., 2023). ProGEN (Madani et al., 2023) uses Transformer (Vaswani et al., 2017) for protein sequence design. ProtGPT2 (Ferruz et al., 2022) deploys a GPT model (Radford et al., 2018) to generate protein sequences. Evodiff (Alamdari et al., 2023) implements protein sequence design through an autoregressive diffusion model. Lutz et al. (2023) utilizes reinforcement learning to design the structure of proteins. RFDiffusion (Watson et al., 2023) uses a diffusion model to generate the structure of proteins and then obtains the corresponding sequences through ProteinMPNN (Dauparas et al., 2022). AlphaProteo (Zambaldi et al., 2024) employs a generator to design high-affinity binders, achieving a high success rate. We focus on the design of antibodies, which are important proteins for disease treatment.

**Antibody design.**  Early antibody design methods relied on Monte Carlo iterations to update and generate antibody sequences and structures through handcrafted functions (Pantazes & Maranas, 2010; Lapidoth et al., 2015; Adolf-Bryfogle et al., 2018; Warszawski et al., 2019; Ruffolo et al., 2021). However, this approach is resource-intensive and typically produces antibodies that conform strictly to the energy function, heavily relying on the design of the energy function itself. Sequence models (Alley et al., 2019; Saka et al., 2021; Shin et al., 2021; Akbar et al., 2022) were widely used in the early stages of deep learning for antibody design. RefineGNN (Jin et al., 2022) was the first to propose the co-design of antibody CDRs and structures, utilizing an autoregressive approach to generate the CDR regions of antibodies. DiffAb (Luo et al., 2022) employs diffusion models to design the sequences and structures of CDR regions for specific antigens. MEAN (Kong et al., 2023a) utilizes graph neural networks to iteratively generate the sequences and structures of antibody CDR regions. dyMEAN (Kong et al., 2023b) further expands the capabilities of MEAN, enabling end-to-end antibody design. However, these methods are limited to the design of CDR regions, specifically the CDR H3 region, while the other regions of the antibody must be predefined, rendering them unsuitable for practical antibody design.

**Generative models.**  Generative models are becoming increasingly popular in the field of biomolecular generation (You et al., 2018; Liu et al., 2018), particularly due to the high quality of samples produced by diffusion models (Song & Ermon, 2019; Ho et al., 2020; Leach et al., 2022; Austin et al., 2021). Consequently, a growing number of studies have introduced diffusion models into biomolecular generation. Many works have successfully utilized diffusion models to design multifunctional proteins (Wu et al., 2022; Trippe et al., 2022; Gao et al., 2023). To address the limitations of inference steps in diffusion models, Song et al. (2023) proposed the consistency model. This model can map any point back to the initial point at a given time step while maintaining high quality. Several studies (Luo et al., 2023; Sauer et al., 2023; Xiao et al., 2023; Wang et al., 2023) have demonstrated the effectiveness of the consistency model. In this work, we utilize the consistency model to accelerate sampling.

## B  DIFFUSION MODELS AND CONSISTENCY MODELS

### B.1  DIFFUSION MODELS

Diffusion models (Sohl-Dickstein et al., 2015; Song & Ermon, 2019; Ho et al., 2020) are a type of generative model that have been successfully applied in various fields, including image generation (Dhariwal & Nichol, 2021; Nichol & Dhariwal, 2021; Rombach et al., 2022), protein design (Anand & Achim, 2022; Trippe et al., 2023; Komorowska et al., 2024), among others. Diffusion models gradually add noise to the data through a forward process until it becomes random noise, and then learn a reversible backward process to progressively recover the original data from the noise.

**Continuous Diffusion**  To model a distribution $p(w)$, an effective method involves first embedding $w$ into a continuous variable $x_0$ using an embedding matrix $U_\theta$ and then adding Gaussian noise.

For the $C_\alpha$ coordinates of antibodies, the values in three-dimensional space are continuous, and therefore a continuous diffusion model can be used to generate the $C_\alpha$ coordinates (Watson et al., 2023). In practical applications, Gaussian noise can be gradually added to the coordinate values until they approach a Gaussian distribution. The prior distribution is set as $\pi(x) = \mathcal{N}(0, I)$, and the forward process is defined by:

$$p(x_t|x_0) = \mathcal{N}(x_t; \sqrt{\bar{\alpha}_t}x_0, (1 - \bar{\alpha}_t)I) \tag{11}$$

where $\bar{\alpha}_t$ ranges from 0 to 1. The values of $\bar{\alpha}_t$ are determined by a predefined noise schedule.

The reverse process aims to learn a function $p_\theta(\hat{w}|x_t, t)$ that can reconstruct the sequence from the noisy data points $x_t$. This is achieved by minimizing the following loss function with respect to $\theta$:

$$L(\theta) = \mathbb{E}_{w_0, t}\left[-\log p_\theta(w_0|x_t)\right], \quad x_t \sim p(x_t|x_0 = U_\theta w_0). \tag{12}$$

Using $p_\theta(\hat{w}|x_t, t)$, we can define the reverse process distribution as follows:

$$p_\theta(x_{t-1}|x_t) = \sum_{\hat{w}} p\left(x_{t-1}|x_t, \hat{x}_0 = U_\theta \hat{w}\right) p(\hat{w}|x_t, t), \tag{13}$$

where $p(x_{t-1}|x_t, x_0)$ is also a Gaussian distribution. During inference, the learned reverse process can be used to transform samples from $\pi(x)$ into samples from the learned distribution $p_\theta(x_0)$. This is done by iteratively sampling from $p_\theta(x_{t-1}|x_t)$ and then sampling $w \sim p_\theta(\hat{w}|x_0, 0)$.

**Discrete Diffusion** As proposed by Austin et al. (2021), for scalar discrete random variables with $K$ categories, denoted as $s_t$ and $s_{t-1}$ where $s_t, s_{t-1} \in \{1, \ldots, K\}$. For antibody sequences, there are 20 classes of amino acids. For a specific antibody sequence, the amino acid type at each position can be regarded as a categorical distribution. For a random variable at a certain position, there are 20 classes, denoted as $s_t, \ldots, s_{t-1} \in \{1, \ldots, 20\}$. The forward transition probabilities can be represented using matrices: $[Q_t]_{ij} = q(s_t = j|s_{t-1} = i)$. Representing $s$ in its one-hot form as a row vector $\mathbf{s}$, the forward transition can be expressed as:

$$q(\mathbf{s}_t|\mathbf{s}_{t-1}) = \text{Cat}(\mathbf{s}_t; \mathbf{p} = \mathbf{s}_{t-1}Q_t), \tag{14}$$

where $\text{Cat}(\mathbf{s}; \mathbf{p})$ denotes a categorical distribution over the one-hot vector $\mathbf{s}$ with probabilities from $\mathbf{p}$. The term $\mathbf{s}_{t-1}Q_t$ represents a row vector-matrix product, with $Q_t$ applied independently to each pixel or token.

Starting from $\mathbf{s}_0$, the $t$-step marginal and posterior at time $t - 1$ are given by:

$$q(\mathbf{s}_t|\mathbf{s}_0) = \text{Cat}(\mathbf{s}_t; p = \mathbf{s}_0\overline{Q}_t), \quad \text{with} \quad \overline{Q}_t = Q_1 Q_2 \cdots Q_t,$$

$$q(\mathbf{s}_{t-1}|\mathbf{s}_t, \mathbf{s}_0) = \frac{q(\mathbf{s}_t|\mathbf{s}_{t-1}, \mathbf{s}_0)q(\mathbf{s}_{t-1}|\mathbf{s}_0)}{q(\mathbf{s}_t|\mathbf{s}_0)} = \text{Cat}(\mathbf{s}_{t-1}; \mathbf{p} = \frac{\mathbf{s}_t Q_t^\top \odot \mathbf{s}_0\overline{Q}_{t-1}}{\mathbf{s}_0\overline{Q}_t\mathbf{s}_t^\top}). \tag{15}$$

**SO(3) Diffusion** For each amino acid direction, the orientation within the local coordinate system of each amino acid can be considered as a continuous value in the SO(3) space, analogous to a uniform distribution on polar coordinates (Leach et al., 2022). Consequently, we can construct both forward and backward diffusion processes on the three-dimensional rotation group SO(3). The forward process diffusing the direction data $o_0$ into pure noise, following the specific formula:

$$q(o_t|o_0) = \text{IGSO}(3)(\lambda(\sqrt{\alpha_t}, o_0), (1 - \alpha_t)) \tag{16}$$

where IGSO(3) denotes the isotropic Gaussian distribution on SO(3), and $\lambda$ represents the scalar product along the geodesic from the identity rotation matrix to $o_0$. Conversely, the backward process is designed to transform noise back into data, guided by the following probability distribution:

$$p(o_{t-1}|o_t, o_0) = \text{IGSO}(3)(\widetilde{\mu}(o_t, o_0), \widetilde{\beta}_t) \tag{17}$$

where $\widetilde{\mu}(o_t, o_0)$ denotes the mean of the backward process, which is calculated as the product of two rotation matrices.

### B.2 Consistency Models

Consistency Models(CMs) (Song et al., 2023; Song & Dhariwal, 2024) leverage the PF-ODE framework to create a direct relationship between data and noise distributions. The primary objective of CMs is to develop a consistency function $f(\mathbf{x}_t, t)$ that effectively transforms a noisy image $\mathbf{x}_t$ back into its clean counterpart $\mathbf{x}_0$, adhering to a boundary condition at $t = 0$. This is accomplished through a specific parameterization:

$$f_\theta(\mathbf{x}_t, t) = c_{\text{skip}}(t)\,\mathbf{x}_t + c_{\text{out}}(t)\,F_\theta(\mathbf{x}_t, t), \tag{18}$$

where the conditions $c_{\text{skip}}(0) = 1$ and $c_{\text{out}}(0) = 0$ ensure compliance with the boundary requirement. During training, the PF-ODE is discretized into $N - 1$ segments and a loss function is minimized to quantify the difference between adjacent points along the sampling path:

$$\arg\min_\theta \mathbb{E}\left[\lambda(t_i)d(f_\theta(\mathbf{x}_{t_{n+1}}, t_{n+1}), f_{\theta^-}(\tilde{\mathbf{x}}_{t_n}, t_n))\right]. \tag{19}$$

In this equation, $d(\cdot, \cdot)$ represents a chosen metric, $f_{\theta^-}$ denotes an exponential moving average of previous outputs, and $\tilde{\mathbf{x}}_{t_n}$ is calculated based on the noise gradient. The selection of the metric and the sampling strategy is vital for effective model training.

## C  Experiment details

### C.1  Dataset

We selected all experimentally determined antibody structures published in the database up to December 31, 2022, as our training set. The final training set consisted of 6,448 antibody-antigen complexes with both heavy and light chains and 1,907 single-chain antibody-antigen complexes. During the training process, we used CD-Hit (Li & Godzik, 2006) to cluster the training set, with each cluster containing antibodies with sequence similarities above 95%, resulting in a total of 2,436 clusters. To ensure the utilization of available data, we randomly sampled one sample from each cluster for training in each epoch. The validation and test sets included antigen-antibody complexes determined experimentally and published between January 1, 2023, and June 30, 2023, and between June 30, 2023, and December 30, 2023, respectively. More details can be found at Appendix C.3. We removed sequences that were similar to those in the training set to eliminate redundancy in the data, ensuring a fair evaluation. This process resulted in 101 validation samples and 60 test samples, both of which were completely unrelated to the training set. The validation set was used for hyperparameter tuning and model selection, while the test set, named SAb-23H2-Ab, was utilized for subsequent performance evaluation. We also utilized a nanobody dataset released in the second half of 2023 to construct a test set for nanobodies, referred to as SAb-23H2-Nano.

### C.2  Model details

#### C.2.1  Inter-chain Feature Embedding Module

The input to the Inter-chain Feature Embedding Module includes a list of chain information (chn_infos), an asymmetric ID vector (asym_id), the number of embedding dimensions ( $n\_$dims), and the maximum relative index (ridx_max). The algorithm begins by initializing a linear layer with an input dimension of $2\times$ ridx_ max $+3$ and an output dimension of $n\_$dims. Next, an asymmetric matrix is computed based on the asym_id and converted into a floating-point tensor. An index vector is then generated to calculate the relative index matrix. After being trimmed and adjusted, the final relative index matrix is obtained and one-hot encoded. Subsequently, the asymmetric tensor and the one-hot encoded tensor are concatenated along the last dimension and processed through the linear layer to produce an updated feature tensor. Finally, the algorithm outputs this updated tensor for subsequent feature processing. This process effectively integrates the relative positions of the chains and asymmetric information, providing the model with rich contextual information.

## C.2.2 STRUCTURE ENCODER

The structural module consists of two components: one for encoding spatial information and the other for encoding the interaction information between the antigen and antibody based on epitope information. For spatial encoding, the pairwise distances between $C_\alpha$ atoms in each sample are calculated using the three-dimensional coordinates of each alpha carbon atom, resulting in the generation of the corresponding distance mask. The distance values in the distance tensor are then mapped to their respective bin indices, effectively discretizing the distances into equidistant variable indices. By embedding these indices and incorporating the distance mask, the structural encoding is obtained. Epitope information encoding is utilized to generate embedded representations of the interaction and contact map features between antigens and antibodies. Initially, the input epitope information is mapped to a specified feature dimension through a linear layer, followed by further projection to the target dimension. During the transmission of epitope information, the module pre-processes the input epitope data to generate the corresponding feature representations. Specifically, an appropriate feature generation method is selected based on the dimensionality of the input features, ensuring that the type of features is consistent with the weight type of the projection layer. The preprocessed features are then passed through the embedding layer and the projection layer to produce the final feature embedding. Through these steps, the epitope information encoding module effectively processes and transforms the input epitope data, generating features that enable the model to recognize epitope locations, thereby facilitating the effective generation of antibodies targeting specific epitopes, as illustrated in the Figure 8.

## C.3 TRAINING DETAILS

The training process of IgGM is divided into two stages, following the concept of curriculum learning. Initially, we train the model for structural design, we use the Adam (Loshchilov & Hutter, 2017) optimizer and set the batch size of the training process to 32. We also maintain an EMA (Exponential Moving Average) decay of 0.999 for the model parameters and evaluate the model, selecting the best TM-Score on the validation subset as the optimal model. Given an antigen-antibody complex, we perturb the antibody structure to introduce noise, and then have the model reconstruct the perturbed antibody structure. This process lasted for 5 days on 8 A100 GPUs. Once the first-stage model has converged, we use the parameters from the first-stage training to proceed with the second stage. In the second stage, we perturb the sequence and structure of the antibody's CDR regions and have the model reconstruct the perturbed antibody. This perturbation is aimed at introducing greater complexity and variability into the training data, thereby challenging the model to generalize better to unseen data. It is noteworthy that during the second phase of training, we employed a mixed training approach. In the model training process, we assigned probabilities of $4 : 2 : 2 : 2$ for the model to design CDR H3, CDR H, all CDRs, and to refrain from sequence design, respectively. This process also lasted for 5 days on 8 A100 GPUs. Throughout both stages, we use self-conditioning (Chen et al.) to enhance the stability of the training. This self-conditioning technique involves feeding the model with additional information derived from the original unperturbed structure, which helps the model to better learn the underlying patterns and regularities in the data. Ensuring that the model learns robustly and can generalize well to new and challenging data. The two-stage training process allows the model to first learn basic patterns and then progressively build on that knowledge to handle more complex scenarios. After training the generative model, we distill the consistency model using the method proposed by Song et al. (2023). For specific details, please refer to Song et al. (2023).

Table 4: Different sample steps for IgGM.

| Method | AAR↑ CDR3 | DockQ↑ | iRMS↓ | LRMS↓ | SR↑ |
|---|---|---|---|---|---|
| Step=1 | 0.362 | 0.240 | 6.474 | 19.694 | 0.383 |
| Step=2 | 0.363 | 0.244 | 6.570 | 19.616 | 0.383 |
| Step=5 | 0.363 | 0.242 | 6.610 | 19.932 | 0.400 |
| Step=10 | 0.360 | 0.246 | 6.579 | 19.678 | 0.433 |
| Step=20 | 0.361 | 0.232 | 6.840 | 20.980 | 0.400 |
| Step=50 | 0.348 | 0.225 | 6.809 | 20.905 | 0.383 |

## C.4 EVALUATION METRICS

For the sequence portion, we employ metrics that have been widely used in previous work (Luo et al., 2022; Jin et al., 2021; 2022; Kong et al., 2023a;b): AAR, the amino acid recovery rate represents the proportion of similarity between the designed antibody sequence and the actual antibody sequence. A higher value indicates that the model has a greater ability to generate a specific antibody.

For the evaluation of antibody structure design, we employ several established protein structure assessment metrics, including the RMSD, TM-Score (Zhang & Skolnick, 2004), GDT-TS (Zemla, 2003), DockQ (Basu & Wallner, 2016), and SR.

- Root Mean Square Deviation (RMSD): This metric obtained by calculating the mean of the squared differences between the coordinates of aligned CDR H3 backbone atoms and then taking the square root. The CDR H3 region is the most flexible, making it more challenging to predict.

- TM-Score (Template Modeling Score): This metric measures the structural similarity between the predicted antibody structure and a reference structure. A TM-Score of 1 indicates an exact match between the two structures, while a score closer to 0 indicates a poor match, a high TM-Score suggests that the predicted structure closely resembles the native structure.

- GDT-TS (Global Distance Test-Total Score): This metric provides an overall assessment of the model's accuracy by comparing the predicted structure to the native structure based on the global distance test. It takes into account both the accuracy of the model's predictions and the conformational similarity to the native structure. A higher GDT-TS score indicates a better match between the predicted structure and the native structure, suggesting a higher quality design.

- DockQ: This metric is specifically designed for evaluating the quality of protein-protein docking predictions. It assesses the interface complementarity and the conformational accuracy of the predicted complex structure. A high DockQ value indicates that the predicted interface is likely to be functional and stable, suggesting a well-designed multi-chain interface.

$$\text{DockQ} = \frac{\text{Fnat} + \text{RMS}_{\text{scaled}}(\text{LRMS}, d_1) + \text{RMS}_{\text{scaled}}(\text{iRMS}, d_2)}{3}, \tag{20}$$

where $\text{RMS}_{\text{scaled}}$ represents the scaled RMS deviation corresponding to either LRMS or iRMS, $d_i$ is a scaling factor, $d_1$ is used for LRMS, and $d_2$ is used for iRMS. Fnat is defined as the fraction of native contacts retained in the predicted complex interface.

- SR (Success Rate): Indicates that the quality of the designed multi-chain interface positioning is within an acceptable range when DockQ is greater than 0.23. High SR indicates that more structure of antibodies is good.

## D THE OBJECTIVE OF IGGM

For amino acid sequences, given that there are only 20 types of amino acids, different amino acids exhibit similar backbone atoms. We treat sequence recovery as a classification problem. We employ cross-entropy loss to guide the model in learning the correct sequence.

When it comes to the structure of antibodies, as previously mentioned in Appendix B.1, our objective is to recover the spatial coordinates of alpha carbons and the orientations of backbone atoms. To this end, we utilize the residue Frame Mean Squared Error (FMSE) loss, which has been demonstrated to be effective in (Watson et al., 2023). This loss function is specifically designed to measure the discrepancy between the predicted and actual frames of protein residues, which are essential for accurately modeling the three-dimensional structure of antibodies. We have further developed an enhanced loss function termed inter-chain Frame Mean Squared Error (iFMSE) loss. This loss function is specifically tailored to impose constraints on the differences between different chains within the antibody structure. The iFMSE loss is designed to ensure that the model accurately captures the relative orientations and positions of the various chains that make up the antibody, which is crucial for maintaining the integrity of the quaternary structure. Additionally, we employ the inter-residue distance and angle metrics to reconstruct the orientations of backbone atoms. By

incorporating these loss functions, our model is trained to not only accurately predict the amino acid sequences but also to reconstruct the intricate three-dimensional structures of antibodies, thereby enhancing the potential of our approach in the fields of structural biology and antibody modeling.

The overall objective of the model is to recover the correct sequence and structure from the denoised antibody data. The overall loss function can be represented as follows:

$$\mathcal{L} = \mathcal{L}_{\text{srcv}} + \mathcal{L}_{\text{geo}} + \mathcal{L}_{\text{Frame}} + \mathcal{L}_{\text{iFrame}} + 0.02\mathcal{L}_{\text{viol}}. \tag{21}$$

- **Amino-acid Sequence Recovery Loss** $\mathcal{L}_{\textbf{srcv}}$ is designed to supervise the model to recover the amino-acid sequence $s_i$ at the position $i$. A total 20 classes for common amino acid types are considered. Sequence embedding $\{s_i\}$ are linearly projected into the output classes and scored with the cross-entropy loss:

$$\mathcal{L}_{\text{srcv}} = -\frac{1}{\sum_{i \in \text{design}}} \sum_{c=1}^{20} y_i^c \log p_i^c, \tag{22}$$

  where $p_i^c$ are predicted class probabilities, $y_i^c$ are one-hot encoded ground-truth values, and averaging across the masked positions.

- **Inter-residue Geometric Loss** $\mathcal{L}_{\textbf{geo}}$ is designed to provide more direct supervision in the following stack. Four auxiliary heads, implemented as feed-forward layers, are added to the top of the final pair features for predicting inter-residue distances and angles, as described in trRosetta (Yang et al., 2020). These include:

  - $D_{ij}$: Distance between $C_\beta$ and $C_\beta'$
  - $\Omega_{ij}$: Dihedral angle formed by $C_\alpha$, $C_\beta$, $C_\beta'$, and $C_\alpha'$
  - $\Theta_{ij}$: Dihedral angle formed by $N$, $C_\alpha$, $C_\beta$, and $C_\beta'$
  - $\Phi_{ij}$: Planar angle formed by $C_\alpha$, $C_\beta$, and $C_\beta'$

  Each prediction head outputs a probabilistic estimation of the aforementioned distance and angles, denoted as $\text{logits}_{ij}^d$, $\text{logits}_{ij}^\omega$, $\text{logits}_{ij}^\theta$, and $\text{logits}_{ij}^\varphi$. The cross-entropy loss is calculated for each term and summed up to form the final inter-residue geometric loss:

$$\mathcal{L}_{\text{geo}} = \sum_{ij} \text{CE}\left(\text{logits}_{ij}^d; D_{ij}\right) + \text{CE}\left(\text{logits}_{ij}^\omega; \Omega_{ij}\right) + \text{CE}\left(\text{logits}_{ij}^\Theta; \theta_{ij}\right) + \text{CE}\left(\text{logits}_{ij}^\Phi; \varphi_{ij}\right) \tag{23}$$

- **Residue Frame MSE Loss** is designed to provide more direct supervision in the predict module to recover the structure of antibody. The formula can be expressed as follows:

$$L_{\text{Frame}} = \frac{1}{\sum_{i=0}^{I-1} \gamma^i} \sum_{i=1}^{I} \gamma^{I-i} d_{\text{Frame}}(x^{(0)}, \hat{x}^{(0),i})^2. \tag{24}$$

  where $d_{\text{Frame}}(x^{(0)}, \hat{x}^{(0)})$ represents the distance about both rotation and translation, and it can be expressed in the following form:

$$d_{\text{Frame}}(x^{(0)}, \hat{x}^{(0)}) = \sqrt{\frac{1}{L} \sum_{l=1}^{L} \left( w_{\text{t}} \min\left( \left\| z_l^{(0)} - \hat{z}_l^{(0)} \right\|^2, d_{\text{clamp}} \right)^2 + w_{\text{r}} \left\| I_3 - \hat{r}_l^{(0)\top} r_l^{(0)} \right\|_F^2 \right)}, \tag{25}$$

  where $w_{\text{t}}$ and $w_{\text{r}}$ are weights on the translation and rotation distances, and $d_{\text{clamp}}$ is a maximum distance to avoid the numerical overflow.

- **Interface Residue Frame MSE Loss** is designed to provide more direct supervision in the predict module to recover the structure of antibody between interchain. The formula can be expressed as follows:

$$L_{\text{iFrame}} = \frac{1}{\sum_{i=0}^{I-1} \gamma^i} \sum_{i=1}^{I} \gamma^{I-i} d_{\text{iFrame}}(x^{(0)}, \hat{x}^{(0),i})^2, \tag{26}$$

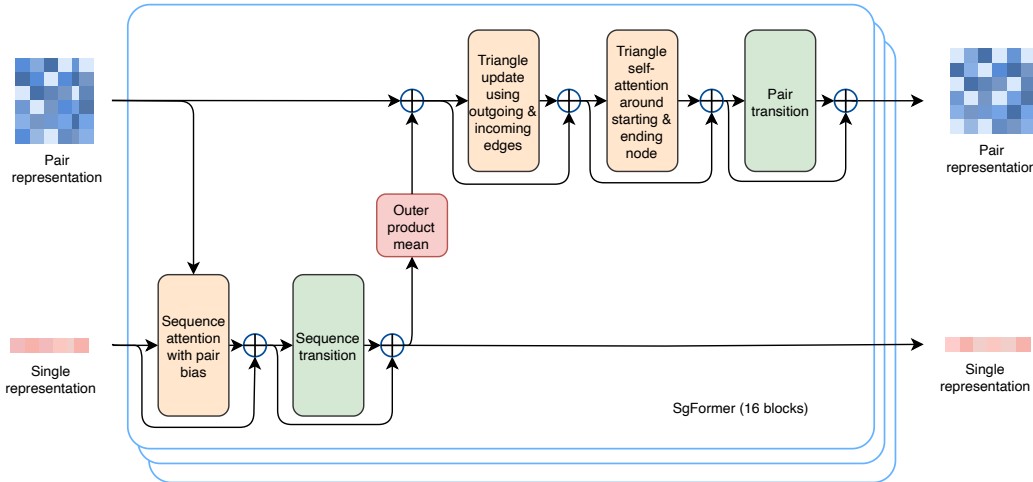

Figure 5: The SgFormer blocks update the single and pair representations through sequence attention and triangle update module.

where $d_{\text{iFrame}}(x^{(0)}, \hat{x}^{(0)})$ represents the distance about both rotation and translation, and it can be expressed in the following form:

$$d_{\text{iFrame}}(x^{(0)}, \hat{x}^{(0)}) = \sqrt{\frac{1}{L_{\text{interface}}} \sum_{l=1}^{L_{\text{interface}}} \left( w_{\text{t}} \min\left( \left\| z_l^{(0)} - \hat{z}_l^{(0)} \right\|^2, d_{\text{clamp}} \right)^2 + w_{\text{r}} \left\| I_3 - \hat{r}_l^{(0)\top} r_l^{(0)} \right\|_F^2 \right)},$$

(27)

where $w_{\text{t}}$ and $w_{\text{r}}$ are weights on the translation and rotation distances, and $d_{\text{clamp}}$ is a maximum distance to avoid the numerical overflow, $L_{\text{interface}}$ represents the amino acid sequence of the contact region.

- **Structure violation loss $\mathcal{L}_{\text{viol}}$:** Similar to AlphaFold2 (Jumper et al., 2021), we introduce penalty terms for incorrect peptide bond length and angles, as well as steric clashes between non-bonded atoms. For multimer structure prediction, we do not penalize the bond length and angle between the last residue in the heavy chain and the first residue in the light chain, since there is no peptide bond between them. Besides, we normalize the steric clash loss by the number of non-bonded atom pairs in clash to stabilize the model optimization, as suggested in AlphaFold-Multimer (Evans et al., 2021).

$$\mathcal{L}_{\text{viol}} = \mathcal{L}_{\text{bond-length}} + \mathcal{L}_{\text{bond-angle}} + \mathcal{L}_{\text{clash}}$$

(28)

## E    ABLATION STUDY

We conducted ablation experiments on parts of the model and evaluated it on the test set, as shown in Table 5. The removal of the two-stage training resulted in a significant decline in all metrics, particularly with the SR dropping to 0, indicating that the two-stage training is crucial for model performance. The absence of epitope information led to a marked deterioration in the DockQ, iRMS, and LRMS metrics, highlighting the importance of epitope information on the docking quality and bias of the model. After removing ESM-PPI, there was a slight decrease in AAR and DockQ, while iRMS and LRMS increased, suggesting that ESM-PPI has a certain impact on the bias of the interface and ligand. The removal of mixed training also resulted in a slight decline in AAR and DockQ, with increases in iRMS and LRMS, indicating that mixed training has a notable effect on the overall performance of the model. Overall, the various components and training strategies of the IgGM model significantly influence its performance, with the two-stage training and epitope information being particularly critical for the docking quality and success rate of the model.

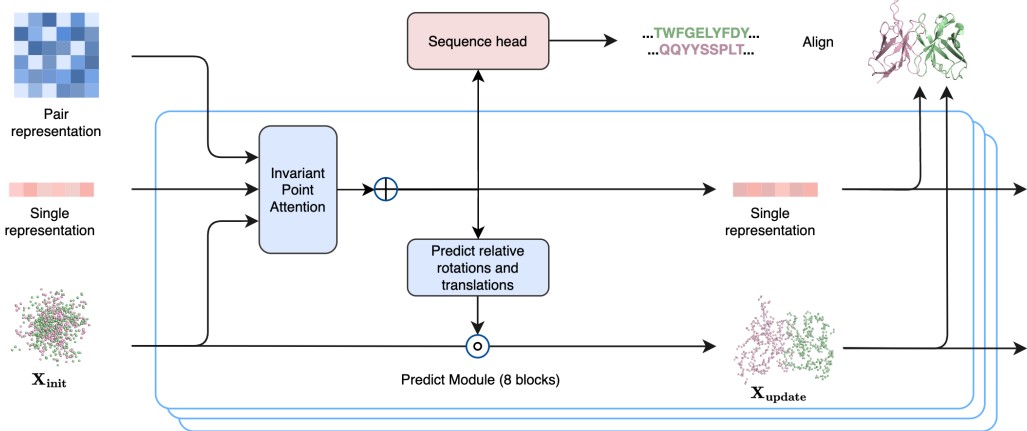

Figure 6: Antibody Sequence-Structure Prediction Module: The features input into a prediction module that includes an IPA module and a Sequence head to predict the sequence and structure of the antibody.

Table 5: Ablations performance for IgGM.

| Method | AAR↑ CDR3 | DockQ↑ | iRMS↓ | LRMS↓ | SR↑ |
|---|---|---|---|---|---|
| IgGM | 0.360 | 0.246 | 6.579 | 19.678 | 0.433 |
| w/o two stage training | 0.160 | 0.072 | 10.260 | 30.961 | 0.000 |
| w/o epotipe | 0.326 | 0.069 | 14.609 | 36.967 | 0.050 |
| w/o ESM-PPI | 0.322 | 0.233 | 7.444 | 20.996 | 0.426 |
| w/o mixed training | 0.334 | 0.231 | 7.524 | 22.713 | 0.350 |

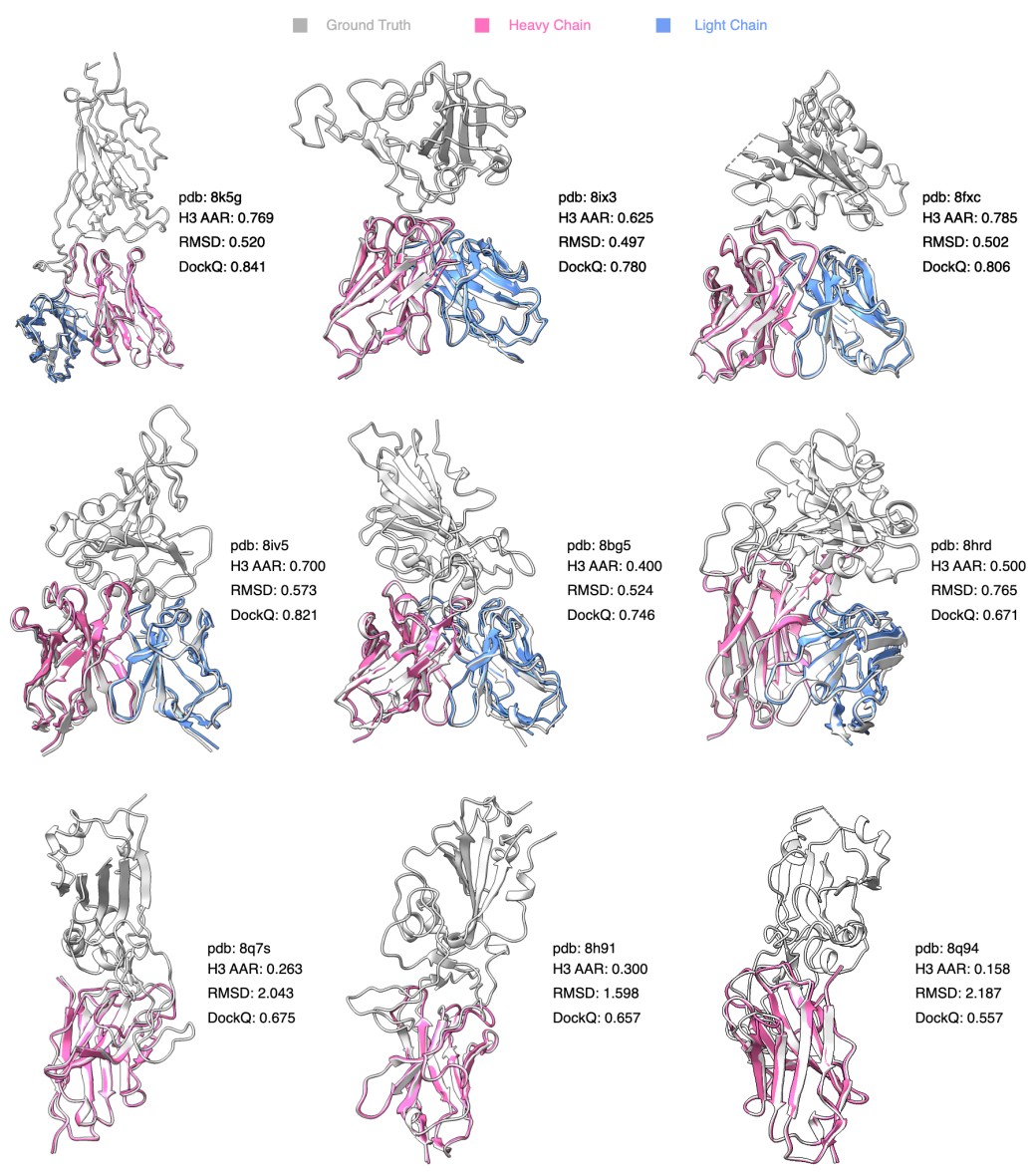

Figure 7: Samples of generated antibodies and nanobodies by IgGM.

## F EXAMPLES

We present additional examples of the de novo designs generated by our IgGM in Figure 7. As depicted in the Figure 8, by employing IgGM, we are capable of generating specific antibodies tailored to various lengths and different epitopes. Moreover, as illustrated in Figure 9, IgGM has the ability to rectify the incorrect positions predicted by AlphaFold3, bringing them close to the epitope.

## G LIMITATIONS AND FUTURE DIRECTIONS

With the advancement of antibody design methods, the feasibility of AI-designed antibodies has become a reality (Khetan et al., 2022). However, existing experimental metrics are insufficient to effectively evaluate whether the antibodies designed by these models can truly bind to the antigen.

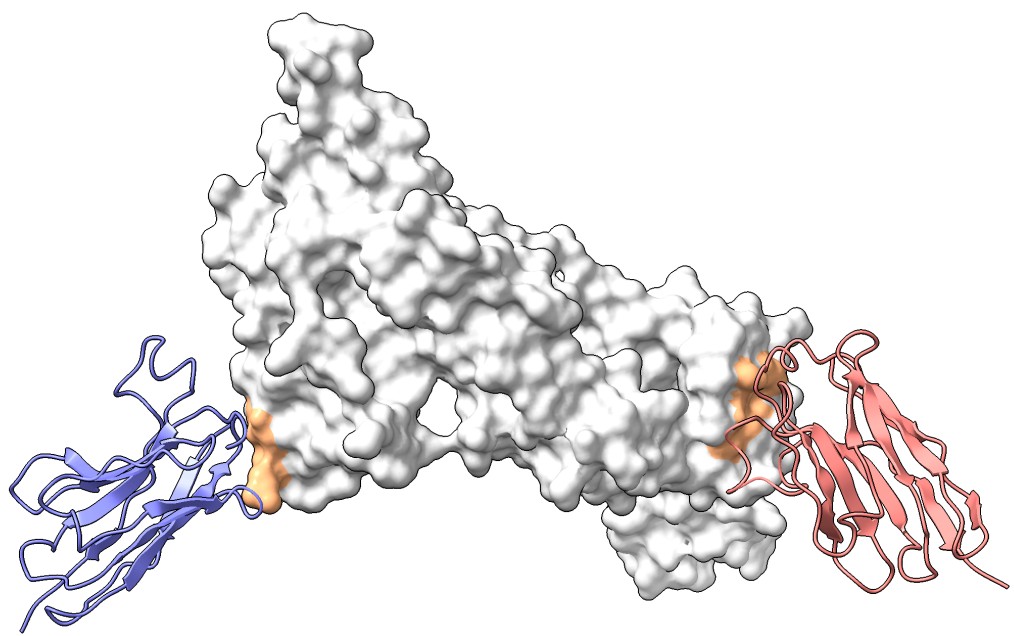

Figure 8: IgGM designs nanobodies targeting different epitopes of (PDB ID: 7MMN), with different colors representing the various designed nanobodies. The original binding entity of this antigen are antibodies.

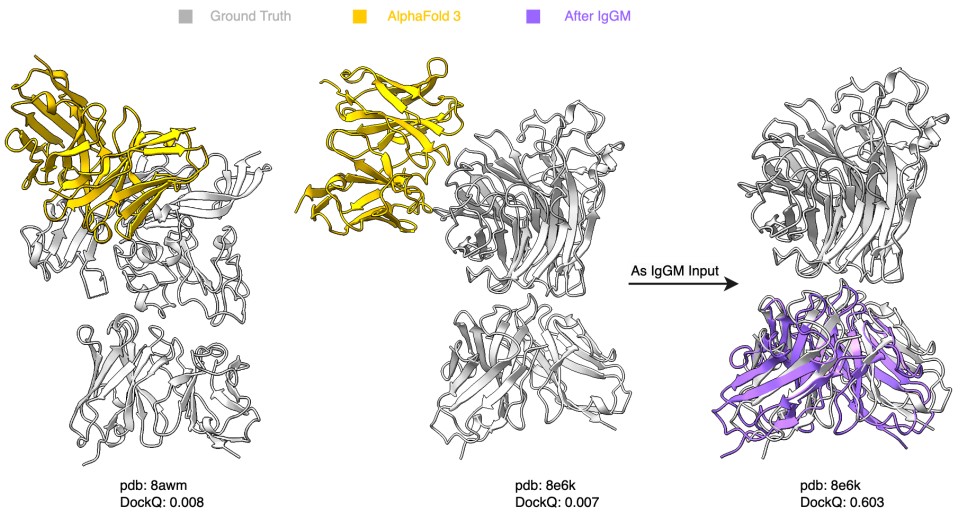

Figure 9: Examples of deviations in the complex structures predicted by AlphaFold3 can be corrected using IgGM, resulting in more accurate structures.

A key reason for this limitation is the significant gap between in silico evaluations and wet lab experimental results. Bridging this gap is crucial, as it would allow in silico evaluations to significantly accelerate the antibody development process. This approach holds promise across all fields of AI-driven drug discovery. Currently, in silico evaluations rely on comparisons with existing antibodies; however, for entirely new antigens, such references are unavailable. Therefore, efficiently and accurately screening for effective antibodies remains a critical research challenge.

Beyond the realm of antibody design, AlphaProteo (Zambaldi et al., 2024) has achieved a high success rate in binder design by utilizing a generator to produce candidate binders, followed by screening with a discriminator. This method has significantly improved success rates. AlphaProteo employs a generator-discriminator approach, where IgGM can serve as the generator. However, there is currently no tool capable of effectively identifying binding antibodies, and the unique characteristics of antibodies make the development of relevant tools even more challenging. Nevertheless, this remains one of the future research directions: designing an efficient discriminator for screening and evaluating the affinity and specificity of antibody candidates. By effectively filtering out inefficient or unsuitable antibodies during the computational simulation phase, this approach could greatly reduce the number of candidates requiring wet lab validation. This not only enhances the efficiency of antibody design but also accelerates the drug development process and reduces research and development costs. In the future, we aim to develop a discriminator or utilize existing discrimination matrices to perform screening, applying these methods in practical scenarios for wet lab validation.

