# OpenReview forum: "IgGM: A Generative Model for Functional Antibody and Nanobody Design"
_ICLR.cc/2025/Conference — ICLR 2025 Poster_

### Official Review · Reviewer_ar8p · 2024-10-28

**Soundness:** 3
**Presentation:** 3
**Contribution:** 3
**Rating:** 6
**Confidence:** 4

**Summary:**

This paper introduces IgGM, a generative model for designing antibody/antigen structures and sequences for a given antigen. IgGM uses features from a pre-trained language model and a diffusion framework to generate antibodies and nanobodies. The model shows great performance in both structure prediction and antibody design.

**Strengths:**

1. The authors made a comprehensive comparision between IgGM and other baseline methods cross multiple metrics, demonstrating impressive results.
2. This paper is well-structued, with carefully prepared figures that enhance the reader's understanding.

**Weaknesses:**

Major comments:

1. One major limitation is that the generation of antibodies depends on their framework regions, which predetermine the length of CDRs and may not always be available when handling a new antigen. Since the optimal CDR lengths are not known a priori, the authors should demonstrate the capability of IgGM in designing antibodies with varying CDR lengths.

2. However, in line 147, the authors state that their method "...enables the design of the whole antibody structure even without experimental structures." This claim requires clarification, as the results presented only demonstrate the ability to design CDRs.

3. I suggest that the authors should compare their method with other antibody deisgn methods, such as:
 - Co-design models: AbX [1] (very similar framework with IgGM)
 - Structure-only models: RFdiffusion for antibody design [2]
 - Sequence-only models: AbLang [3] and IgLM [4]

Minor comments:

1. In figure  6, "Aligin" should be "Align"

References

[1] Zhu T, Ren M, Zhang H. Antibody Design Using a Score-based Diffusion Model Guided by Evolutionary, Physical and Geometric Constraints[C]//Forty-first International Conference on Machine Learning.

[2] Bennett N R, Watson J L, Ragotte R J, et al. Atomically accurate de novo design of single-domain antibodies[J]. bioRxiv, 2024.

[3] Olsen T H, Moal I H, Deane C M. AbLang: an antibody language model for completing antibody sequences[J]. Bioinformatics Advances, 2022, 2(1): vbac046.

[4] Shuai R W, Ruffolo J A, Gray J J. IgLM: Infilling language modeling for antibody sequence design[J]. Cell Systems, 2023, 14(11): 979-989. e4.

**Questions:**

1. How is epitope information encoded in the model, and to what extent can the model follow the guidance of a specific epitope?

---

> ### Author Response · Authors · 2024-11-22
> **Response to Reviewer ar8p (1/3): Response to the Weaknesses Part(Major1, 2)**
>
> Dear reviewer ar8p,
>
> We express our gratitude for your diligent review and the valuable efforts you have made to assist us in enhancing the manuscript.
>
> * **W1:** One major limitation is that the generation of antibodies depends on their framework regions, which predetermine the length of CDRs and may not always be available when handling a new antigen. Since the optimal CDR lengths are not known a priori, the authors should demonstrate the capability of IgGM in designing antibodies with varying CDR lengths.
>
> The fixed length is intended to facilitate comparisons with target antibodies. In reality, our model only requires the input of the antibody framework region sequence to design the CDRs and the overall structure of the antibody. The framework region is generally more conserved, primarily serving to support the loops of the CDR regions. In practical design, specific framework regions are often selected to ensure the stability of certain physicochemical properties [1] and to facilitate humanization of the antibody (to prevent rejection by the human body). \
> \
> In IgGM, users can specify the length of the CDRs during input to design CDRs of varying lengths. We have included instructions (Example 5) for this operation in the anonymous repository (https://anonymous.4open.science/r/IgGM), and you can run the code to verify this functionality. Additionally, in the updated version of the manuscript, we have added schematic diagrams for various scenarios, which you can review directly.
>
> * **W2:** However, in line 147, the authors state that their method "...enables the design of the whole antibody structure even without experimental structures." This claim requires clarification, as the results presented only demonstrate the ability to design CDRs.
>
> You can refer to W1, where it is stated that IgGM can design the entire structure of an antibody by simply providing a framework region sequence (**No antibody structures are required**) with favorable properties.

---

> ### Author Response · Authors · 2024-11-22
> **Response to Reviewer ar8p (2/3): Response to the Weaknesses Part(Major3, Minor1)**
>
> * **W3:** I suggest that the authors should compare their method with other antibody deisgn methods,
>
> We have conducted a comparison of these methods, as summarized in the table below. For AbX [1], this method represents an improvement over the DiffAb definition problem; however, it is limited to designing CDR regions and cannot predict the entire structure of an antibody, nor can it facilitate the design of entirely new antibodies. In the case of RFDiffusion, its original version [5] does not support antibody design. Our attempts to use the Binder approach did not yield a folded antibody structure. As for antibody version [2], we are currently unable to test it due to the lack of released code.\
> \
> Additionally, we found that language models [3][4], having been trained on large datasets, achieve a relatively high level of sequence recovery. However, these models can only be used for antibody sequence completion and are not capable of generating sequences specific to particular antigens, rendering them impractical for real-world applications. Nevertheless, their performance raises important questions about how to effectively utilize these extensive sequences to enhance the generation of high-quality antibodies, which may be a key direction for future research.\
> \
> Furthermore, it has been observed that there is considerable variation in the execution methods and inputs of these methods. Consequently, we have revised the input formats of these approaches to facilitate execution in a single-step process. These modifications will be incorporated into the repository for forthcoming community evaluation.
>
> | Model   |       | AbLang    | IgLM  |  AbX(AF3) | dyMEAN |    IgGM    | IgGM    (AF3) |
> |---------|:-----:|-----------|-------|:---------:|--------|:----------:|:-------------:|
> | AAR     |   L1  | **0.758** | 0.694 |   0.655   | 0.633  |   0\.750   |     0\.737    |
> |         |   L2  | 0.717     | 0.639 |   0.464   | 0.634  |   0\.743   |     0\.735    |
> |         |   L3  | **0.656** | 0.594 |   0.598   | 0.570  |   0\.635   |     0\.602    |
> |         |   H1  | 0.735     | 0.650 | **0.752** | 0.742  |   0\.740   |     0\.739    |
> |         |   H2  | **0.691** | 0.578 |   0.677   | 0.627  |   0\.644   |     0\.639    |
> |         |   H3  | 0.352     | 0.278 |   0.338   | 0.294  | **0\.360** |     0\.330    |
> | RMSD    |   L1  | -         | -     |   0.815   | 0.864  | **0\.589** |     0\.659    |
> |         |   L2  | -         | -     |   0.491   | 0.481  | **0\.378** |     0\.395    |
> |         |   L3  | -         | -     |   1.029   | 0.941  | **0\.847** |     0\.903    |
> |         |   H1  | -         | -     |   0.635   | 0.633  | **0\.555** |     0\.590    |
> |         |   H2  | -         | -     |   0.625   | 0.705  | **0\.486** |     0\.566    |
> |         |   H3  | -         | -     |   2.753   | 2.454  | **2\.131** |     2\.155    |
> | Docking | DockQ | -         | -     |   0.208   | 0.079  |   0\.246   |   **0\.326**  |
> |         |  iRMS | -         | -     |   7.217   | 9.698  |   6\.579   |   **4\.030**  |
> |         |  LRMS | -         | -     |   24.207  | 28.764 |   19\.678  |  **11\.229**  |
> |         |   SR  | -         | -     |   0.360   | 0.049  |   0\.433   |   **0\.627**  |
>
> \setkeys{Gin}{width=1.25\linewidth}
> ![This is alt text to describe my image.](review_pic/add_results.png ")
>
> [1] Zhu, Tian, Milong Ren, and Haicang Zhang. "Antibody Design Using a Score-based Diffusion Model Guided by Evolutionary, Physical and Geometric Constraints." Forty-first International Conference on Machine Learning. \
> [2] Bennett, Nathaniel R., et al. "Atomically accurate de novo design of single-domain antibodies." bioRxiv (2024). \
> [3]Olsen, Tobias H., Iain H. Moal, and Charlotte M. Deane. "AbLang: an antibody language model for completing antibody sequences." Bioinformatics Advances 2.1 (2022): vbac046.\
> [4]Shuai, Richard W., Jeffrey A. Ruffolo, and Jeffrey J. Gray. "IgLM: Infilling language modeling for antibody sequence design." Cell Systems 14.11 (2023): 979-989.\
> [5]Watson, Joseph L., et al. "De novo design of protein structure and function with RFdiffusion." Nature 620.7976 (2023): 1089-1100.
>
> * **W4:** In figure 6, "Aligin" should be "Align"
>
> Thank you for pointing out this issue. We have made updates in the new version.

---

> ### Author Response · Authors · 2024-11-22
> **Response to Reviewer ar8p (3/3):  Response to the Questions Part**
>
> * **Q1:** How is epitope information encoded in the model, and to what extent can the model follow the guidance of a specific epitope?
>
> The epitope information encoding method utilizes a special input encoding (We have provided clarification in the updated paper) that is mapped to feature space through a Multi-Layer Perceptron (MLP). For further details, you can refer to the Python code in the repository. Additionally, IgGM allows for the specification of different positions to generate antibodies targeting specific epitopes. In the Figure 8, we have included the design of nanobodies targeting two epitopes of (PDB ID: 7MMN, the original binding entity of this antigen are antibodies.), with the framework region selected as follows:
> 'QVQLVESGGGLVQPGGSLRLSCAASXXXXXXXXXXXLGWFRQAPGQGLEAVAAXXXXXXXXYYADSVKGRFTISRDNSKNTLYLQMNSLRAEDTAVYYCXXXXXXXXXXXXXXXXXXWGQGTLVTVSS'[6], you can also execute anonymous code (https://anonymous.4open.science/r/IgGM) for generation purposes. Specifically, you can refer to Example 4, which demonstrates how to design antibodies targeting specific epitopes by specifying different positions.
>
> [6] Vincke, Cecile, et al. "General strategy to humanize a camelid single-domain antibody and identification of a universal humanized nanobody scaffold." Journal of Biological Chemistry 284.5 (2009): 3273-3284.

---

> ### Comment · Reviewer_ar8p · 2024-11-22
>
> I appreciate the authors' efforts in addressing the concerns raised through detailed experiments and responses. The additional experiments demonstrate the strong performance of IgGM compared to other structure-based antibody design methods. Therefore, I have decided to update my rating to 6.

---

> > ### Author Response · Authors · 2024-11-25
> >
> > Thank you for reading our rebuttal and for your supportive words. We are pleased to hear that we have addressed all your concerns. Once again, we would like to express our gratitude for your insightful comments, which have been truly inspiring and have significantly contributed to the enhancement of our paper. If you have any further questions or would like to discuss anything in more detail, please feel free to reach out to us.
> >
> > Sincerely,\
> > Authors

---

### Official Review · Reviewer_R8Qh · 2024-10-30

**Soundness:** 3
**Presentation:** 2
**Contribution:** 2
**Rating:** 5
**Confidence:** 4

**Summary:**

This paper introduces IgGM, a diffusion model for simultaneously generating the structure and sequence of antibody CDR regions, as well as the structure of antibody FR regions. By employing a two-stage training approach (folding -> design), IgGM is capable of conducting antibody design and docking tasks, and achieves commendable performance.

**Strengths:**

1. Compared to most existing antibody design works, this study undertakes the design of the antibody CDR region without providing the antibody framework, consistent with dyMEAN. This setting is closer to real-world requirements and poses a greater challenge.

2. The two-stage training approach brings a significant performance improvement.

3. Additionally, targeted designs focusing on inter-chain interactions were conducted.

**Weaknesses:**

1. It seems that the work does not fundamentally differ from DiffAb in terms of diffusion algorithm, and it bears some resemblance to ESMFold in model design.

2. Having only five samples for each example is somewhat limited.

3. The main paper does not provide an explanation of the diffusion algorithm, nor does it detail the representation of proteins and the prior distribution associated with each modality.

**Questions:**

1. In line 278, the probabilities of 4:2:2:2 are assigned to the model to design CDR H3, CDR H, and all CDR. So, do you mean CDR-H3 only for 4/10, all CDR-Hs for 2/10, and all CDRs for 2/10 ?

2. I am confused as to why you ultimately trained a consistency model. I comprehend that the key advantage of a consistency model lies in accelerating the generation process, enabling completion in fewer steps. However, in a scientific problem such as antibody design, is acceleration truly significant? Related to this, you have employed 10 different sampling steps in the appendix, but aside from the success rate (SR), the other four metrics seem to decline with the increase in the number of steps. Even SR shows a decline after surpassing 10 steps. These experimental results differ from what we might anticipate; I would appreciate a detailed study and explanation. Additionally, how does the original model perform without using the consistency model?

3. Some physical metrics (like energy) are needed for a more refined evaluation.

4. The model name (DiffIg) in Figure 4 has not been updated.

5. The experiment in Section 4.2 seems to be the most important one in this paper, but I am somewhat unclear about the experimental setup here.

    5.1. 'Structure prediction⇒docking⇒CDR generation⇒side-chain packing' is not the process of dyMEAN. On the contrary, this is the 'Existing Works Pipeline' shown in Figure 1 to highlight the end-to-end approach of dyMEAN.

    5.2. In the stage of structure prediction, how is the sequence of the CDR part defined? Are random sequences used, or are special symbols like [MASK] employed?

    5.3. The AAR of dyMEAN on CDR-H3 is much lower than reported in the literature. Although the training and test data used in this paper differ from the original dyMEAN article, such a significant disparity needs to be explained (43.65%->29.4%).

    5.4. In both Section 4.1 and 4.2, there is a version of IgGM that uses AF3-predicted structures for initialization, making the prior distribution inconsistent with N(0, I). How does IgGM predict the structure in this scenario? Was an additional IgGM trained with AF3 predicted structures as prior? If it's the latter, how were the noise addition and denoising processes conducted?"

6. Finally, for any researcher working on AI-driven antibody design, aligning in silico evaluation with wet-lab experimental results is an essential consideration. Unfortunately, I did not find any discussion related to this in the main paper. If IgGM were to be eventually validated experimentally, do you believe the advantages of IgGM over other methods would still persist? Also, what improvements could be made?

---

> ### Author Response · Authors · 2024-11-22
> **Response to Reviewer R8Qh (1/4): Response to the Weaknesses Part**
>
> Dear reviewer R8Qh,
>
> We express our gratitude for your diligent review and the valuable efforts you have made to assist us in enhancing the manuscript.
>
> * **W1:** It seems that the work does not fundamentally differ from DiffAb in terms of diffusion algorithm, and it bears some resemblance to ESMFold in model design.
>
> DiffAb first integrates three types of diffusion and demonstrates their effectiveness. However, for DiffAb, this work serves more as a filling function, specifically filling in the CDR regions given the structure of the antigen-antibody complex. ESMFold performs well in the prediction of monomeric protein structures. However, ESMFold cannot directly predict the structures of complexes. Although it can indirectly predict them through linkers, its performance is suboptimal, particularly for antigen-antibody complex structures.
>
> Our goal is not to design an entirely new universal model, but to focus on the field of antibodies and address the challenges that current models cannot solve. Our model can design antibodies that bind near epitopes given a specific antigen, while also predicting high-quality complex structures.  Additionally, IgGM achieves structural prediction, antibody design, and nanobody design simultaneously without the need for retraining, whereas other methods require retraining for specific tasks. And allowing it to design antibodies from scratch while also utilizing structures predicted by advanced tools like AlphaFold 3 as initialization to enhance structural accuracy—capabilities that are not present in other methods. We hope that IgGM will contribute to advancements in antibody drug design.
>
> * **W2:** Having only five samples for each example is somewhat limited.
>
> Due to the limitations of AlphaFold 3 server, we generated five structures for each antigen-antibody complex as initialization. To ensure a fair comparison, we selected these five structures and also conducted 100 generations, with the results summarized as follows. It can be observed that the results from the five structures are not significantly different from those obtained from the 100 generations.
>
> | Model   |       | dyMEAN (ori.) | dyMEAN (new.) | IgGM(ori.) | IgGM (new.) |
> | :------ | :---- | :------------ | :------------ | :--------- | :---------- |
> | AAR     | L1    | 0\.633        | 0\.627        | 0\.750     | 0\.745      |
> |         | L2    | 0\.634        | 0\.620        | 0\.743     | 0\.738      |
> |         | L3    | 0\.570        | 0\.553        | 0\.635     | 0\.624      |
> |         | H1    | 0\.742        | 0\.748        | 0\.740     | 0\.737      |
> |         | H2    | 0\.627        | 0\.623        | 0\.644     | 0\.635      |
> |         | H3    | 0\.294        | 0\.291        | 0\.360     | 0\.355      |
> | RMSD    | L1    | 0\.864        | 0\.737        | 0\.589     | 0\.614      |
> |         | L2    | 0\.481        | 0\.455        | 0\.378     | 0\.375      |
> |         | L3    | 0\.941        | 0\.873        | 0\.847     | 0\.852      |
> |         | H1    | 0\.633        | 0\.624        | 0\.555     | 0\.552      |
> |         | H2    | 0\.705        | 0\.686        | 0\.486     | 0\.516      |
> |         | H3    | 2\.454        | 2\.350        | 2\.131     | 2\.143      |
> | Docking | DockQ | 0\.079        | 0\.078        | 0\.246     | 0\.236      |
> |         | iRMS  | 9\.698        | 8\.000        | 6\.579     | 5\.420      |
> |         | LRMS  | 28\.764       | 29\.164       | 19\.678    | 20\.241     |
> |         | SR    | 0\.049        | 0\.048        | 0\.433     | 0\.425      |
>
> * **W3:** The main paper does not provide an explanation of the diffusion algorithm, nor does it detail the representation of proteins and the prior distribution associated with each modality.
>
> Due to the constraints of the main text's length, we have provided relevant explanations of the algorithm in Appendix B.

---

> ### Author Response · Authors · 2024-11-22
> **Response to Reviewer R8Qh (2/4): Response to the Questions Part(Q1,2,3,4)**
>
> * **Q1:** In line 278, the probabilities of 4:2:2:2 are assigned to the model to design CDR H3, CDR H, and all CDR. So, do you mean CDR-H3 only for 4/10, all CDR-Hs for 2/10, and all CDRs for 2/10 ?
>
> Here, we conducted a mixed ratio training with a ratio of 4:2:2:2, randomly allowing the model to perform [CDR H3 sequences-whole structure], [CDR H sequences-whole structure], [CDR sequences-whole structure], and [whole structure] tasks. We employed random numbers to handle different components, enabling the model to possess the capability to design various regions of antibodies and nanobodies without the need for retraining for different tasks, as required by other methods. We have updated the text with a more precise description of this approach. This is a form of multi-task mixed training, where the proportions are related to the difficulty of the problems, as CDRH3 is the most flexible and challenging region to design for antibodies.
>
> * **Q2-1:** I am confused as to why you ultimately trained a consistency model. I comprehend that the key advantage of a consistency model lies in accelerating the generation process, enabling completion in fewer steps. However, in a scientific problem such as antibody design, is acceleration truly significant?
>
> This is a profound and insightful question that touches upon the trade-off between efficiency and accuracy in scientific research. One of the advantages of the consistency model is its ability to accelerate the generation process. For antibodies, the high specificity often results in a low success rate in wet lab experiments [1]. After generating a batch of candidate antibodies, further screening of these candidates is necessary in subsequent experiments. Therefore, the rapid and large-scale generation of candidate antibodies is crucial. Another advantage of the consistency model is its capability to perform various edits and applications in zero-shot settings without explicit training. This is because the consistency model can map any point at any time step back to the starting point of the trajectory, allowing IgGM not only to generate antibodies from noise but also to effectively integrate with other tools, such as AlphaFold 3, to achieve higher quality antibody generation.
>
> [1] Bennett, Nathaniel R., et al. "Atomically accurate de novo design of single-domain antibodies." bioRxiv (2024).
>
> * **Q2-2:** Related to this, you have employed 10 different sampling steps in the appendix, but aside from the success rate (SR), the other four metrics seem to decline with the increase in the number of steps. Even SR shows a decline after surpassing 10 steps. These experimental results differ from what we might anticipate; I would appreciate a detailed study and explanation. Additionally, how does the original model perform without using the consistency model?
>
> Regarding the metrics, unlike image-related metrics (e.g., FID and IS, which primarily measure distribution consistency), the relevant metrics for antibodies are calculated and compared precisely against the Ground Truth. Even slight deviations can lead to significant changes in results. We tested various step counts for IgGM, and the results showed slight differences between different step counts. We selected 10 steps because, for proteins, their functions are primarily determined by their structures, and the accuracy of the structure is more valuable for assessing antibody functionality. This choice represents a balance between speed and accuracy. Additionally, given the variability in results across different step counts, different sampling steps can be employed to further enhance the diversity of the model's outcomes.\
> \
> We present the results obtained using only the diffusion model below.
>
> * **Q3:** Some physical metrics (like energy) are needed for a more refined evaluation.
>
> Energy is not a particularly reliable metric for evaluating antibody design, especially in cases where structural inaccuracies are present. And we have included an assessment of these results below. The calculation of this energy involves first relaxing the structure to its minimum energy using the Rosetta software package, followed by the use of the InterfaceAnalyzer tool within the package to compute the binding energy. IMP refers to the percentage of designed antibodies that exhibit lower (better) binding energy (∆G) compared to the original antibody. In calculating these energies, we also relaxed the original antibody to obtain its minimum energy, which was then used to compute the IMP. Experimental results indicate that IgGM outperformed existing methods.
>
> | Model  | DiffAb | dyMEAN | IgGM   | IgGM(AF3) |
> | :----- | :----- | :----- | :----- | :-------- |
> | IMP(\%) | 1.25  | 6.88  | 16.21 | 17.86    |
>
>
> * **Q4:** The model name (DiffIg) in Figure 4 has not been updated.
>
> Thank you for pointing out this issue. We have made updates in the new version.

---

> ### Author Response · Authors · 2024-11-22
> **Response to Reviewer R8Qh (3/4): Response to the Questions Part(Q5)**
>
> * **Q5.1:** 'Structure prediction⇒docking⇒CDR generation⇒side-chain packing' is not the process of dyMEAN. On the contrary, this is the 'Existing Works Pipeline' shown in Figure 1 to highlight the end-to-end approach of dyMEAN.
>
> Apologies for the oversight. What I intended to convey is that we used the pipeline for testing other methods from dyMEAN as Pipeline 1. We have revised this description in the newly uploaded version. In line 415, we emphasized the end-to-end nature of dyMEAN.
>
> * **Q5.2:** In the stage of structure prediction, how is the sequence of the CDR part defined? Are random sequences used, or are special symbols like [MASK] employed?
>
> In structural prediction tasks, the common approach is to input the raw sequence, similar to AlphaFold. In design tasks, both the structure and sequence are generated together, with the initial input consisting of randomly sampled sequences and structures. IgGM can simultaneously design the sequence and the final structure of antibodies. We also tested the use of [MASK] and found that this method was less effective than using randomly sampled sequences.
>
> * **Q5.3:** The AAR of dyMEAN on CDR-H3 is much lower than reported in the literature. Although the training and test data used in this paper differ from the original dyMEAN article, such a significant disparity needs to be explained (43.65\%→29.4\%).
>
>  We analyzed the proportion of amino acids generated by the dyMEAN method and found that dyMEAN tends to produce Y and G. For the dyMEAN test set (RAbD), using the paradigm 'ARDG ∗ ∗ ∗ DY where most ∗ are Y' yields an AAR of 39.61\%. As explained in the response to reviewer wZ8n Q2, this trend does not hold for the new test set, where the application of this paradigm results in only a 20.18\% sequence recovery rate. Our analysis indicates that dyMEAN's preference for generating Y (see Figure 7) may be the reason for the decline in its performance. We have addressed this phenomenon in our updated article.
>
> * **Q5.4:** In both Section 4.1 and 4.2, there is a version of IgGM that uses AF3-predicted structures for initialization, making the prior distribution inconsistent with N(0, I). How does IgGM predict the structure in this scenario? Was an additional IgGM trained with AF3 predicted structures as prior? If it's the latter, how were the noise addition and denoising processes conducted?"
>
> We addressed this in Q2, highlighting that this is indeed an advantage of the consistency model. We have provided further explanations in the manuscript to help readers gain a clearer understanding of this point. Furthermore, under conditions of extremely low probability, it is still possible to sample structures predicted by AlphaFold 3 from a distribution of N(0, I).

---

> ### Author Response · Authors · 2024-11-22
> **Response to Reviewer R8Qh (4/4): Response to the Questions Part(Q6)**
>
> * **Q6:** Finally, for any researcher working on AI-driven antibody design, aligning in silico evaluation with wet-lab experimental results is an essential consideration. Unfortunately, I did not find any discussion related to this in the main paper. If IgGM were to be eventually validated experimentally, do you believe the advantages of IgGM over other methods would still persist? Also, what improvements could be made?
>
> Aligning in silico evaluation with wet-lab experimental results is crucial for AI-driven drug discovery, and the significance of this issue determines the pace of development in AI pharmaceuticals. Effective in silico evaluation can significantly shorten the drug development process. However, there remains a gap between in silico evaluations and wet-lab experiments, particularly for antibodies, which are proteins that bind specifically to certain antigens. A specific antibody will only function against its corresponding antigen. Current in silico evaluations primarily focus on comparing the similarity to existing antibodies, but for entirely new antigens, these metrics are not applicable for assessment.
>
> While [1] has demonstrated the feasibility of de novo antibody design through wet experiments. Additionally, there are approaches that involve the maturation and modification of existing antibodies to design antibodies with higher affinity.[3] However, the success rate of all these methods is relatively low. Therefore, the most accurate assessment of AI designs still relies on wet-lab experiments. This highlights the need for more reliable in silico evaluation methods that can predict the efficacy of antibodies against new antigens.
>
> Beyond the field of antibodies, DeepMind's publication on AlphaProteo [4] has shown that AI can be used to design high-affinity binders, AlphaProteo employs a generator-discriminator approach, where IgGM can serve as the generator. In the future, we aim to develop a discriminator or utilize existing discrimination matrices to perform screening, applying these methods in practical scenarios for wet lab validation. This approach is highly relevant for antibody screening and suggests a potential pathway for improving in silico evaluations.
>
> However, there is currently no method that can accurately assess antibody affinity. We are confident that the emergence of precise affinity discriminators will further facilitate the alignment of in silico evaluations with wet-lab experiments. Developing such discriminators will involve creating models that can accurately predict the binding affinity of antibodies to their target antigens, potentially using advanced machine learning techniques and large datasets of antibody-antigen interactions.
>
> We will also explore this aspect in our future work to establish a comprehensive antibody design and screening process. This will include integrating more sophisticated in silico evaluation methods with experimental validation to create a more efficient and reliable pipeline for antibody discovery.
>
> [1] Bennett, Nathaniel R., et al. "Atomically accurate de novo design of single-domain antibodies." bioRxiv (2024).\
> [3] Cai, Huiyu, et al. "Pretrainable geometric graph neural network for antibody affinity maturation." Nature communications 15.1 (2024): 7785.\
> [4] Zambaldi, Vinicius, et al. "De novo design of high-affinity protein binders with AlphaProteo." arXiv preprint arXiv:2409.08022 (2024).

---

> > ### Comment · Reviewer_R8Qh · 2024-11-24
> >
> > Thanks for the detailed rebuttal. My major questions have been solved and I increased the score to 5.

---

> > > ### Author Response · Authors · 2024-11-25
> > >
> > > Thank you for reading our rebuttal and for your supportive words. We are pleased to hear that we have addressed your concerns. Once again, we would like to express our gratitude for your insightful comments, which have been truly inspiring and have significantly contributed to the enhancement of our paper. If you have any further questions or would like to discuss anything in more detail, please feel free to reach out to us.
> > >
> > > Sincerely,\
> > > Authors

---

### Official Review · Reviewer_wZ8n · 2024-11-12

**Soundness:** 3
**Presentation:** 3
**Contribution:** 2
**Rating:** 5
**Confidence:** 4

**Summary:**

This paper introduces IgGM, a generative model designed for creating functional antibody and nanobody structures. It integrates sequence and structural generation using a multi-level network approach comprising a pre-trained language model, feature encoder, and prediction module. Experimental results demonstrate its applicability in antibody and nanobody design tasks.

**Strengths:**

1. IgGM successfully establishes an antibody co-design pipeline that integrates several key elements previously deemed essential by the research community.
2. IgGM shows promising docking success rates over AF3, suggesting that it effectively captures essential antibody-antigen interaction patterns.

**Weaknesses:**

1. Although the IgGM framework demonstrates its effectiveness on various antibody design tasks, it relies heavily on components and algorithms established in prior work, limiting the originality and impact of its contributions to the field.
2. The study lacks direct comparisons with RFDiffusion while they perform similar tasks
3. The authors didn't report the variance/robustness of their performance.

**Questions:**

1.  It would be valuable to explore why AlphaFold 3 (AF3) shows stronger performance in structure prediction than in docking-related metrics, perhaps due to limitations in capturing finer antigen-binding dynamics.
2. Why did the authors opt not to use RAbD as the test set, as has been customary in prior studies?

---

> ### Author Response · Authors · 2024-11-22
> **Response to Reviewer wZ8n (1/2): Response to the Weaknesses Part**
>
> Dear reviewer wZ8n,
>
> We express our gratitude for your diligent review and the valuable efforts you have made to assist us in enhancing the manuscript.
>
> * **W1:** Although the IgGM framework demonstrates its effectiveness on various antibody design tasks, it relies heavily on components and algorithms established in prior work, limiting the originality and impact of its contributions to the field.
>
> Hello, previous work, such as AlphaFold and ESMFold, has demonstrated excellent performance in protein structure prediction. However, its performance in antigen-antibody interactions has been relatively poor, and existing methods are unable to generate entirely new antibodies. We have expanded upon existing methods to achieve high-quality antibody design and complex structure prediction. Our goal is not to create a completely new model, but to focus on the field of antibodies and address the challenges that current models cannot solve. Our model can design antibodies that bind near epitopes given a specific antigen, while also predicting high-quality complex structures. Furthermore, it is capable of designing nanobodies, a novel type of antibody that is widely used in the market today—capabilities that existing models cannot provide. We hope that IgGM will contribute to advancements in antibody drug design.
>
> * **W2:** The study lacks direct comparisons with RFDiffusion while they perform similar tasks.
>
> RFDiffusion [1] is not capable of antibody design. The team conducted fine-tuning of RFDiffusion specifically for antibodies in [2]; however, the code for that article has not been released. We attempted to run several cases using the BinderDesign approach from [1], but we were unable to obtain the structures of the antibodies.
>
> [1] Watson, Joseph L., et al. "De novo design of protein structure and function with RFdiffusion." Nature 620.7976 (2023): 1089-1100. \
> [2] Bennett, Nathaniel R., et al. "Atomically accurate de novo design of single-domain antibodies." bioRxiv (2024).
>
> * **W3:** The authors didn't report the variance/robustness of their performance.
>
> We calculated the standard deviation of the models, which is presented in the table below. Overall, the IgGM model exhibits a lower standard deviation in most cases, particularly in terms of structural aspects, indicating that its results are relatively stable.
>
> | Model   |       | dyMEAN (avg.) | dyMEAN (std.) | IgGM(avg.) | IgGM (std.) | IgGM    (AF3/avg.) | IgGM (AF3/std.) |
> | :------ | :---- | :------------ | :------------ | :--------- | :---------- | :----------------- | :------------- |
> | AAR     | L1    | 0\.633        | 0\.141        | **0\.750** | 0\.170      | 0\.737             | 0\.172         |
> |         | L2    | 0\.634        | 0\.189        | **0\.743** | 0\.195      | 0\.735             | 0\.196         |
> |         | L3    | 0\.570        | 0\.193        | **0\.635** | 0\.201      | 0\.602             | 0\.189         |
> |         | H1    | **0\.742**    | 0\.714        | 0\.740     | 0\.175      | 0\.739             | 0\.187         |
> |         | H2    | 0\.627        | 0\.194        | **0\.644** | 0\.238      | 0\.639             | 0\.235         |
> |         | H3    | 0\.294        | 0\.155        | **0\.360** | 0\.232      | 0\.330             | 0\.215         |
> | RMSD    | L1    | 0\.864        | 0\.399        | **0\.589** | 0\.494      | 0\.659             | 0\.528         |
> |         | L2    | 0\.481        | 0\.389        | **0\.378** | 0\.308      | 0\.395             | 0\.249         |
> |         | L3    | 0\.941        | 0\.546        | **0\.847** | 0\.609      | 0\.903             | 0\.636         |
> |         | H1    | 0\.633        | 0\.446        | **0\.555** | 0\.475      | 0\.590             | 0\.540         |
> |         | H2    | 0\.705        | 0\.453        | **0\.486** | 0\.412      | 0\.566             | 0\.454         |
> |         | H3    | 2\.454        | 1\.228        | **2\.131** | 1\.373      | 2\.155             | 1\.156         |
> | Docking | DockQ | 0\.079        | 0\.075        | 0\.246     | 0\.230      | **0\.326**         | 0\.221         |
> |         | iRMS  | 9\.698        | 3\.894        | 6\.579     | 5\.420      | **4\.030**         | 2\.810         |
> |         | LRMS  | 28\.764       | 14\.829       | 19\.678    | 18\.177     | **11\.229**        | 9\.954         |
> |         | SR    | 0\.049        | -             | 0\.433     | -           | **0\.627**         | -              |

---

> ### Author Response · Authors · 2024-11-22
> **Response to Reviewer wZ8n (2/2): Response to the Questions Part**
>
> * **Q1:** It would be valuable to explore why AlphaFold 3 (AF3) shows stronger performance in structure prediction than in docking-related metrics, perhaps due to limitations in capturing finer antigen-binding dynamics.
>
> AlphaFold 3 demonstrates strong performance in the structural prediction of antibodies, which we believe is attributed to its training on a vast dataset that includes nearly all proteins and other relevant data. This rich dataset enables AlphaFold 3 to effectively capture the interactions between amino acid atoms. Additionally, AlphaFold 3 has shown that different types of data can mutually enhance each other, leading to a more refined understanding of internal structures. However, when it comes to metrics related to complexes, the number of available structures is actually limited, particularly for antigen-antibody complexes. In addition, the lack of effective co-evolution information for antigens and antibodies makes it difficult to obtain MSA, which can also lead to poorer performance. This scarcity may explain why AlphaFold 3's performance on antigen-antibody complexes is not as robust as that for monomers.
>
> * **Q2:** Why did the authors opt not to use RAbD as the test set, as has been customary in prior studies?
>
> We have considered this from two perspectives. First, regarding RAbD, the data was released in 2017 [3] and does not include the latest high-quality data. As described in dyMEAN [4], using the pattern 'ARDG ∗ ∗ ∗ DY where most ∗ are Y' achieves an AAR of 39.61\% on the RAbD test set for CDR3. This raises concerns about the validity of testing on the original dataset. Therefore, we constructed a new test set using the most recent data to incorporate the latest information and to avoid such biases. In the new dataset, the phenomenon of achieving high sequence recovery rates through fixed paradigms on RAbD will not occur. Using this paradigm results in a sequence recovery rate of only 20.18\%, which effectively mitigates evaluation errors caused by data bias. Using this paradigm results in a sequence recovery rate of only 20.18\%, which effectively mitigates evaluation errors caused by data bias.
>
> [3] Adolf-Bryfogle, Jared, et al. "RosettaAntibodyDesign (RAbD): A general framework for computational antibody design." PLoS computational biology 14.4 (2018): e1006112. \
> [4] Kong, Xiangzhe, Wenbing Huang, and Yang Liu. "End-to-End Full-Atom Antibody Design." International Conference on Machine Learning. PMLR, 2023.

---

> > ### Comment · Reviewer_wZ8n · 2024-11-25
> > **Response to Reviewer's Rebuttal**
> >
> > I appreciate the authors' efforts in addressing the concerns. Some of my concerns have been resolved, but I did not see further analysis regarding the differences between structure prediction and docking-related metrics. Therefore, I maintain my original opinion.

---

> ### Author Response · Authors · 2024-11-25
> **Reply to the differences between structure prediction and docking-related metrics**
>
> Thank you for your response. We acknowledge that there may have been some discrepancies in our previous discussion regarding the differences between AlphaFold3 in terms of structural predictions and docking. We have conducted further analysis below in hopes of addressing your concerns:
>
> First, we would like to clarify the evaluation areas related to the relevant metrics:
> - **Structure-related metrics**: These indicators measure the similarity of the antibody structure itself, reflecting whether the structure predicted by the structural prediction method aligns with the actual structure.
> - **Docking-related metrics**: These indicators assess the accuracy of the docking between the antigen and the antibody. This set of metrics depends solely on the accuracy of the structural prediction method in determining the relative positions of the antigen and antibody.
> - The region where the antigen and antibody bind is referred to as the **epitope**, which is the pathogenic area of the antigen that the antibody will bind to.
>
> Next, we analyze the performance of AlphaFold3:
> - **Structural Predictions**: AlphaFold3 performs well in predicting antibody structures, as it is currently the most advanced protein structure prediction method. This method has been trained on various protein and molecular datasets, enabling AlphaFold3 to learn the interactions between atoms, which significantly aids in accurately predicting the internal structure of proteins. This is a key reason for AlphaFold3's superior performance in structural predictions.
> - **Docking-related Metrics**: The performance primarily depends on the accuracy of the relative positioning of the antigen and antibody.
>   - For the complex structures predicted by AlphaFold3, there are instances where the binding positions deviate from experimental locations, particularly evident in the incorrect prediction of the epitope's position. This indicates that AlphaFold3 still has limitations in capturing the interactions between the antigen and antibody.
>   - Additionally, for antibody-antigen complexes involving specific epitopes, AlphaFold3 does not support the input of relevant information, which somewhat restricts its performance in docking.
>   - We have updated the manuscript and included examples of the deviations in AlphaFold3 predictions in the appendix (Figure 9), where you can observe significant discrepancies between the predicted antibody positions and the actual positions.
>   - Furthermore, after correction using IgGM, the structures predicted by AlphaFold3 can be adjusted to more accurate positions, with the DockQ score improving from 0.007 to 0.603, as also illustrated in Figure 9.
>
> If you have any further questions or would like to discuss anything in more detail, please feel free to reach out to us.

---

### Official Review · Reviewer_VRyT · 2024-11-13

**Soundness:** 2
**Presentation:** 3
**Contribution:** 4
**Rating:** 8
**Confidence:** 4

**Summary:**

The article discusses the challenges in practical applications where obtaining the structures of antigens and antibodies (including the framework region) is often unfeasible. To address this, it introduces an end-to-end algorithm called IgGM, which simultaneously predicts the sequences and structures of the CDR regions, performs docking based on the epitope, and predicts the structure of the antigen-antibody complex. Additionally, the article employs a two-stage training method to enhance the model's performance in predicting both structures and sequences.

**Strengths:**

The innovation of the article lies in further extending the scope of co-design. The model can not only design amino acids in the CDR region, but also complete antigen docking, which is unprecedented. IgGM not only has excellent performance in predicting antibody structure, especially in the CDR region, but also outperforms existing algorithms in docking. IgGM not only shows SOTA performance on conventional antibodies, but also performs well on nanobodies, further demonstrating the application value of IgGM.

**Weaknesses:**

1. The core architecture of the algorithm is very similar to AF2, and the accuracy of structural prediction seems to be attributed to AF2. However, the innovation in this part is not strong enough.
2. The article did not elaborate on the introduction and explanation of the Inter chain Feature Embedding Module and Structure Encoder.
3. In line 343, the authors define the success rate as DockQ > 0.23. The authors should either provide a reference to justify the selection of this threshold or elaborate on the rationale behind it.
4. From the overall text, it appears that the length of the antibody CDR regions is also specified; however, the authors do not clarify this in the problem formulation.

**Questions:**

1. It was observed in Table 1 that HDock does not have an asterisk (*) symbol. Does this mean that Hdock did not use the information of the epidemic, resulting in lower indicators such as DockQ?
2. ESM-PPI is a model retrained based on Sabdab data. Will the training data of ESM-PPI appear in the validation set of IgGM? Has this part been considered?
3. As shown in Table 1, IgGM (AF3) uses the structure predicted by AF3 as the initial state. However, the authors do not clearly specify whether the sequence input to AF3 is the ground truth sequence. If it is, this would indirectly leak the answer, leading to inflated metrics for both the subsequent structures and sequences. A more rigorous approach would be to use 'initial state sequence + AF3 (random initial state sequence)' instead of 'random initial state sequence + AF3 (ground truth sequence)'

---

> ### Author Response · Authors · 2024-11-22
> **Response to Reviewer VRyT (1/2): Response to the Weaknesses Part**
>
> Dear reviewer VRyT,
>
> We express our gratitude for your diligent review and the valuable efforts you have made to assist us in enhancing the manuscript.
>
> * **W1:** The core architecture of the algorithm is very similar to AF2, and the accuracy of structural prediction seems to be attributed to AF2. However, the innovation in this part is not strong enough.
>
> The success of AlphaFold 2 in protein structure prediction demonstrates the effectiveness of its architecture for proteins. However, the model performs relatively poorly in predicting antigen-antibody structures, as illustrated in AlphaFold 3 [1] Fig. 1c (right). This indicates that there are still deficiencies in the modeling of antigen-antibody interactions. IgGM incorporates the advantages of AlphaFold2 while further optimizing for antibodies. By fully utilizing features such as epitopes and spatial positioning between antigens and antibodies, IgGM not only achieves structural prediction results that surpass those of AlphaFold2 and even AlphaFold3, but also enables the design of entirely new antibodies that bind to specific antigen epitopes.
>
> [1] Abramson, Josh, et al. "Accurate structure prediction of biomolecular interactions with AlphaFold 3." Nature (2024): 1-3.
>
> * **W2:** The article did not elaborate on the introduction and explanation of the Inter chain Feature Embedding Module and Structure Encoder.
>
> In the methods section of our updated article, we provide a detailed explanation of these modules at Appendix C.2. The relevant code is also open-sourced at https://anonymous.4open.science/r/IgGM, where you can access and review the implementation details.
>
> * **W3:** In line 343, the authors define the success rate as DockQ > 0.23. The authors should either provide a reference to justify the selection of this threshold or elaborate on the rationale behind it.
>
> Thank you for your reminder. We have added the citation in the revised version, with the definition sourced from [2]. The DockQ > 0.23 criterion for success is a relatively general metric that has been utilized in many studies, including the well-known works [1][3].
>
> [1] Abramson, Josh, et al. "Accurate structure prediction of biomolecular interactions with AlphaFold 3." Nature (2024): 1-3.\
> [2] Basu, Sankar, and Björn Wallner. "DockQ: a quality measure for protein-protein docking models." PloS one 11.8 (2016): e0161879.\
> [3] Lin, Zeming, et al. "Evolutionary-scale prediction of atomic-level protein structure with a language model." Science 379.6637 (2023): 1123-1130.
>
> * **W4:** From the overall text, it appears that the length of the antibody CDR regions is also specified; however, the authors do not clarify this in the problem formulation.
>
> In our experiments, we fixed the length of the CDRs for the sake of convenient comparison with existing antibodies. This fixation was solely for effective comparison purposes. In practical applications, you can customize the length of the CDRs. Since IgGM only requires the sequence of the framework region, there is no need to worry about structural conflicts when extending the CDR length. This can be verified in the repository code, where the number of 'X' added to the CDR section indicates the CDR length. We also appreciate your reminder. In our updated article, we have included further discussions and added relevant usage instructions (Example 5) in the repository (https://anonymous.4open.science/r/IgGM).

---

> ### Author Response · Authors · 2024-11-22
> **Response to Reviewer VRyT (2/2): Response to the Questions Part**
>
> * **Q1:** It was observed in Table 1 that HDock does not have an asterisk (*) symbol. Does this mean that Hdock did not use the information of the epidemic, resulting in lower indicators such as DockQ?
>
> Yes, this is due to the limitations of HDock, as the method cannot utilize epitope information. We believe this is the reason for the relatively low performance of this metric and represents a limitation of the method itself, as antibodies typically bind only in proximity to the epitope.
>
> * **Q2** ESM-PPI is a model retrained based on SAbDab data. Will the training data of ESM-PPI appear in the validation set of IgGM? Has this part been considered?
>
> ESM-PPI is essentially a language model that does not utilize data from SAbDab. Furthermore, the training of ESM-PPI is based solely on data available prior to 2022. The validation of IgGM is derived from the latest antibodies developed in the second half of 2023, ensuring that there is no overlap in sequences or issues related to data leakage.
>
> * **Q3:** As shown in Table 1, IgGM (AF3) uses the structure predicted by AF3 as the initial state. However, the authors do not clearly specify whether the sequence input to AF3 is the ground truth sequence. If it is, this would indirectly leak the answer, leading to inflated metrics for both the subsequent structures and sequences. A more rigorous approach would be to use 'initial state sequence + AF3 (random initial state sequence)' instead of 'random initial state sequence + AF3 (ground truth sequence)'
>
> Your reasoning is correct. For a completely novel antigen, we cannot obtain relevant antibody information. In such cases, we need to consider how to further leverage existing. For Table 1, the task is structural prediction, defined as generating the structure of an antibody given its true sequence. In this context, using AlphaFold 3 to predict the structure as the initial state is reasonable. For subsequent design tasks, employing the structures predicted by AlphaFold 3 as initialization facilitates comparison with other methods. In practical design, one can replace or randomly initialize the CDR sequences with alanine (A) as an alternative initialization method. This approach, known as alanine scanning, is commonly used in antibody design and optimization to substitute a specific amino acid.
>
> We replaced all CDR regions with alanine (A) and then used AlphaFold to predict the structure as the initial configuration. The results, presented below, indicate that the performance does not significantly decline compared to the structures predicted using the original sequences; in fact, in some metrics, it even surpasses existing benchmarks. This is an intriguing phenomenon, and we will conduct further analysis on how to leverage a better initial structure to achieve improved outcomes, which may also be a valuable direction for research.
>
> | Model   |       | IgGM       | IgGM    (AF3) | IgGM(AF3-CDR all A) |
> | :------ | :---- | :--------- | :------------ | :------------------ |
> | AAR     | L1    | **0\.750** | 0\.737        | 0\.736              |
> |         | L2    | **0\.743** | 0\.735        | 0\.736              |
> |         | L3    | **0\.635** | 0\.602        | 0\.604              |
> |         | H1    | **0\.740** | 0\.739        | 0\.733              |
> |         | H2    | 0\.644     | 0\.639        | **0\.656**          |
> |         | H3    | **0\.360** | 0\.330        | 0\.329              |
> | RMSD    | L1    | **0\.589** | 0\.659        | 0\.634              |
> |         | L2    | 0\.378     | 0\.395        | **0\.375**          |
> |         | L3    | **0\.847** | 0\.903        | 0\.887              |
> |         | H1    | **0\.555** | 0\.590        | 0\.586              |
> |         | H2    | **0\.486** | 0\.566        | 0\.563              |
> |         | H3    | 2\.131     | 2\.155        | **2\.115**          |
> | Docking | DockQ | 0\.246     | **0\.326**    | 0\.302              |
> |         | iRMS  | 6\.579     | 4\.030        | **3\.192**          |
> |         | LRMS  | 19\.678    | 11\.229       | **11\.150**         |
> |         | SR    | 0\.433     | **0\.627**    | 0\.576              |

---

### Official Review · Reviewer_nUM9 · 2024-11-14

**Soundness:** 2
**Presentation:** 2
**Contribution:** 2
**Rating:** 5
**Confidence:** 4

**Summary:**

The authors present IgGM, a novel generative model designed for creating antibodies and nanobodies.

**Strengths:**

The authors curated a high-quality dataset that includes recent data from 2023, ensuring the model is trained on up-to-date antibody and nanobody structures.

By employing diffusion models—a powerful approach in generative modeling—the authors introduce a novel solution to antibody and nanobody design. This choice is especially suited to capturing complex structural and functional patterns in biological data. The study includes ablation studies and comparisons with multiple alternative models.

**Weaknesses:**

The manuscript could benefit from a clearer and more detailed explanation of the features and methods. Improved clarity in this section would enhance readers' understanding of the model’s construction and functionality.

While the authors have effectively applied diffusion models, the novelty is somewhat limited since diffusion models have already been established in various generative tasks.

**Questions:**

The authors did not explicitly mention the threshold used for redundancy reduction in the summary provided. This information is typically included in the methods section, where they would specify the sequence identity cutoff or other criteria used to filter out redundant sequences.

While the authors have shared the test data, any plans for sharing the training data have not been clearly outlined.

---

> ### Author Response · Authors · 2024-11-22
> **Response to Reviewer nUM9**
>
> Dear reviewer nUM9,
>
> We express our gratitude for your diligent review and the valuable efforts you have made to assist us in enhancing the manuscript.
>
> * **W1:** The manuscript could benefit from a clearer and more detailed explanation of the features and methods.
>
> Thank you for your feedback. We have provided further explanations in this section to facilitate readers' understanding.
>
> * **W2:** While the authors have effectively applied diffusion models, the novelty is somewhat limited since diffusion models have already been established in various generative tasks.
>
> Although diffusion models have demonstrated effectiveness in various applications, such as predicting the structures of antigen-antibody complexes (e.g., AlphaFold3) and designing antibody sequences (e.g., AntiBARTy), there is currently no method that can simultaneously design entirely new antibodies and predict the structures of the resulting complexes. IgGM leverages diffusion models while incorporating a consistency model to accelerate the generation of a large number of candidate antibodies. Unlike other models, IgGM is capable of de novo designing antibodies that bind to antigen epitopes, whereas existing methods primarily focus on filling in the CDR3 region. Additionally, IgGM can achieve high-quality structural predictions, a capability that existing models do not possess.
>
> * **Q1:** The authors did not explicitly mention the threshold used for redundancy reduction in the summary provided.
>
> The data processing section is included in Appendix C.1, where we provide a detailed description of the data processing workflow and the construction of the dataset. Relevant explanations are also presented in that section. We have also updated the relevant explanations in the main text to guide interested readers for further exploration.
>
> * **Q2:** While the authors have shared the test data, any plans for sharing the training data have not been clearly outlined.
>
> The dataset has been constructed utilizing the open-source database SAbDab. The original dataset is accessible at https://opig.stats.ox.ac.uk/webapps/sabdab-sabpred/sabdab. Furthermore, our processed data will be made publicly available to facilitate and encourage further research in this domain.

---

### Author Response · Authors · 2024-11-27
**A Summary of Rebuttal**

We thank all reviewers for their comprehensive reviews and insightful suggestions!

Our work focuses on tackling the challenges of AI-designed antibodies in real-world scenarios, aiming to create new antibody designs and extend these capabilities to nanobodies, which have a wide range of applications.

Based on the reviewers' feedback, we have made several revisions and added more experiments to address reviewers' concerns, including:

- We've added more details about the models and technical aspects (Reviewer nUM9, Reviewer R8Qh, Reviewer ar8p) and fixed several errors to make it easier for readers to understand.

- We've included more examples to show how IgGM works with different CDR region lengths (Reviewer ar8p, Reviewer VRyT) and its ability to bind antibodies near specific epitopes (Reviewer ar8p). You can find relevant use cases in our anonymous code repository (https://anonymous.4open.science/r/IgGM), and we plan to organize and release the training data for researchers soon (Reviewer nUM9).

- We've added cases where AlphaFold 3 doesn't predict accurate docking positions (Reviewer wZ8n) to show how IgGM can use epitope information to fix these issues.

- We've explored scenarios where CDR information is unknown (Reviewer VRyT), using alanine substitutions in experiments with AlphaFold 3 predictions, which better reflect real-world design situations.

- We've added a discussion on the differences between dry lab evaluations and wet lab assessments (Reviewer R8Qh) and explored potential future applications and research directions.

- We've included additional comparative methods (Reviewer ar8p) and energy analyses (Reviewer R8Qh), and we plan to integrate various methods into a single evaluation pipeline to help the community assess our work. We hope our research will further advance the practical application of AI-designed antibodies.

We truly appreciate the time and effort each reviewer has put into discussing our manuscript and providing valuable and inspiring feedback.\
If there are any additional questions, we welcome further discussion. Thank you once again for your invaluable contributions.

Best regards,\
Authors

---

### Meta-Review · Area_Chair_YoHY · 2024-12-20

**Metareview:**

The paper introduces IgGM, a generative model for designing antibody and nanobody structures and sequences for a given antigen. IgGM uses features from a pre-trained language model and a diffusion framework to generate antibodies that can effectively bind to specific epitopes on the antigen.

The reviewers have provided constructive feedback, and the authors have adequately addressed the major concerns raised. The paper presents a novel and effective approach for antibody and nanobody design, and the improvements suggested by the reviewers can be incorporated in a revised version to strengthen the submission further.

**Additional Comments On Reviewer Discussion:**

The authors provided a detailed rebuttal addressing the reviewers' concerns. They explained that IgGM can design antibodies with varying CDR lengths by allowing users to specify the CDR lengths during input. The authors clarified that IgGM can design the entire antibody structure, including the framework regions, without requiring experimental structures. They also conducted additional comparisons with other relevant antibody design methods and shared the results. Regarding the encoding of epitope information, the authors provided details on how IgGM can generate antibodies targeting specific epitopes by specifying different positions. Overall, the authors have made significant efforts to address the reviewers' comments and improve the manuscript, which has been positively received by the reviewers.

---

### Decision · Program_Chairs · 2025-01-22

Accept (Poster)